# Accumulator-Aware Post-Training Quantization for Large Language Models

**Ian Colbert**                                          *ian.colbert@amd.com*
*AMD*

**Giuseppe Franco**                                  *giuseppe.franco@amd.com*
*AMD*

**Fabian Grob**                                          *fabian.grob@tum.de*
*TUM\**

**Jinjie Zhang**                                          *zjinjie@amazon.com*
*Amazon†*

**Rayan Saab**                                          *rsaab@ucsd.edu*
*University of California San Diego*

**Reviewed on OpenReview:** *https://openreview.net/forum?id=p6l0579yj7*

## Abstract

When quantizing weights and activations to increasingly narrower representations, the cost of additions begins to dominate that of multiplications in multiply-accumulate (MAC) units. Recent studies show that reducing addition costs via low-precision accumulation improves throughput, power, and area across inference platforms, albeit with an increased risk of overflow. Accumulator-aware quantization research has so far only considered the quantization-aware training (QAT) paradigm, in which models are fine-tuned or trained from scratch with quantization in the loop. As models and datasets continue to grow in size, QAT techniques become increasingly more expensive, which has motivated the recent surge in post-training quantization (PTQ) research. To bridge this gap, we introduce AXE—the first accumulator-aware quantization framework explicitly designed to endow overflow avoidance guarantees to PTQ algorithms. We present theoretical motivation for AXE and demonstrate its flexibility by implementing it on top of two existing algorithms: GPFQ and OPTQ. We design AXE to support multi-stage accumulation, opening the door to full datapath optimization for the first time. We evaluate AXE using recent language generation models; when quantizing Llama3 8B for a 16-bit multi-stage accumulation datapath, AXE maintains up to 98% of the FP16 perplexity, surpassing naïve bit width manipulation by up to 15%.

## 1 Introduction

Neural network quantization is reaching an inflection point. Existing techniques commonly reduce inference costs by restricting the precision of weights and activations to exploit low-precision datapaths in hardware. Although substituting the standard full-precision floating-point operands with low-precision integer counterparts can drastically reduce the cost of multiplications, this only accounts for part of the core multiply-accumulate (MAC) operation; the resulting products are often still accumulated at 32 bits.

Amdahl's Law (Amdahl, 1967) suggests that focusing solely on weights and activations yields diminishing returns. While narrower operand datatypes reduce multiplication costs substantially, they reduce addition

---

*\*Work done while at AMD*
*†Work done while at University of California San Diego*

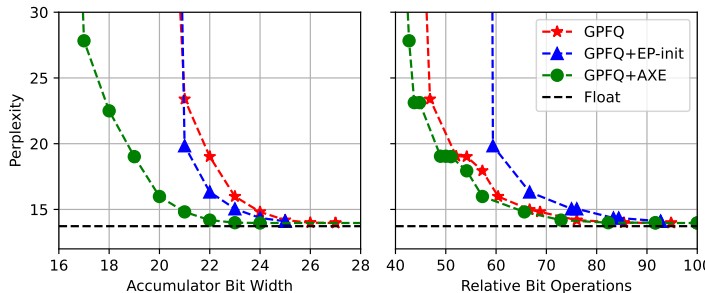

Figure 1: We use GPFQ (Lybrand & Saab, 2021) to quantize SmolLM2-135M (Allal et al., 2024) using naïve bit width manipulations (**red stars**) within the design space described in Section 5. We compare AXE (**green circles**) to EP-init (**blue triangles**) (Colbert et al., 2024) when targeting reduced accumulator bit width. We use Pareto frontiers to visualize the trade-off between WikiText2 (Merity et al., 2016) perplexity and either (left) accumulator bit width or (right) bit operations relative to W8A8 with 32-bit accumulation. Note that our bit operations cost model is highly correlated with relative power savings, as shown in Section 3.1.

costs at a much slower rate. Indeed, recent studies have demonstrated that addition becomes the bottleneck as datatypes shrink, reporting significant benefits when the accumulator precision is restricted during inference. For example, Ni et al. 2020 show that, when constraining operands to 3-bit $\times$ 1-bit multipliers, the cost of 32-bit accumulation consumes nearly 75% of the total power of their MAC unit; they report up to $3\times$ power savings when reducing to 8-bit accumulation. As few-bit integers increase in popularity (Ma et al., 2024; Liu et al., 2025; Zhang et al., 2025b), we expect neural network quantization techniques will need awareness of the accumulator to intentionally address this emerging bottleneck.

Exploiting low-precision accumulation is non-trivial in practice due to three challenges: (1) the benefits of reducing accumulator precision—like those of reducing weight and activation precisions—vary across platforms and workloads, and often depend on specific hardware and software support; (2) even with careful design, reducing accumulator precision exponentially increases the risk of overflow, potentially introducing arithmetic errors that significantly degrade model accuracy (Ni et al., 2020; Colbert et al., 2023); and (3) existing solutions do not scale to modern billion-parameter large language models (LLMs). We focus on the latter two challenges and propose a scalable solution with theoretical justification.

To eliminate the risk of overflow, Colbert et al. 2023 proposed an accumulator-aware quantization paradigm that infuses strict learning constraints informed by theoretical guarantees into quantization-aware training (QAT). The resulting scope of research has since been limited to this QAT setting, where models are trained from scratch or fine-tuned from checkpoints with quantization in the loop (Colbert et al., 2024; Zhang et al., 2025a). With the high training costs of modern deep learning models, it is important to develop methods that are equally as effective in the post-training quantization (PTQ) setting, where pre-trained models are directly quantized and calibrated using relatively modest resources. However, controlling accumulation requirements in such a scenario is non-trivial. To the best of our knowledge, there has been no formal study that explores accumulator-aware quantization in the PTQ setting.

### Contributions.

We provide the first formalization of the accumulator-aware post-training quantization (PTQ) setting and propose AXE as an approximate solution with theoretical justification. AXE infuses overflow avoidance guarantees into layerwise PTQ algorithms that greedily correct quantization error, for example, GPFQ (Lybrand & Saab, 2021) and OPTQ (Frantar et al., 2022). We present AXE as a composition of functions designed to control the dot product ranges throughout error correction, and demonstrate its flexibility by presenting accumulator-aware variants of both GPFQ and OPTQ. Our open-source implementations are made available as part of the Brevitas quantization library v0.12.0[1] (Pappalardo et al., 2025). We evaluate our

---

[1] https://github.com/Xilinx/brevitas/tree/v0.12.0

accumulator-aware variants across pre-trained language generation models and show significant improvements in the trade-off between accumulator bit width and model quality when compared to alternative methods, thereby enabling lower power consumption with better model quality as shown in Figure 1. Unlike prior accumulator-aware QAT methods, which assume a monolithic accumulator, we design AXE to support multi-stage accumulation, which opens the door to datapath optimization and enables large language models (LLMs) for the first time. Indeed, our results show that AXE scales extremely well to billion-parameter language models when targeting multi-stage accumulation. For example, when quantizing Llama3 8B for a 16-bit multi-stage accumulation datapath, AXE maintains up to 98% of the baseline FP16 perplexity.

## 2 Preliminaries

We first introduce our notation. We denote the $K_l$-dimensional input activations to layer $l$ as $\boldsymbol{x}^{(l)} \in \mathbb{R}^{K_l}$, where $\boldsymbol{X}^{(l)} \in \mathbb{R}^{K_l \times D}$ denotes a matrix of $D$ such inputs. The weight matrix for layer $l$ with $K_l$ input neurons and $C_l$ output neurons is similarly denoted as $\boldsymbol{W}^{(l)} \in \mathbb{R}^{C_l \times K_l}$; its quantized counterpart is $\boldsymbol{Q}^{(l)} \in \mathcal{A}_M^{C_l \times K_l}$, where we use $\mathcal{A}_b^{m \times n}$ to denote the space of all $m \times n$ matrices whose elements are part of a fixed $b$-bit alphabet defined by the target quantization space. For example, the alphabet of signed $b$-bit integers is $\mathcal{A}_b := \{k : -2^{b-1} + 1 \leq k \leq 2^{b-1} - 1, k \in \mathbb{Z}\}$, assuming a sign-magnitude representation, where $\mathbb{Z}$ is the space of all scalar integers. For layer $l$, our notation yields $C_l$ independent dot products of depth $K_l$ for each of the $D$ inputs. For clarity, and without loss of generality, we often assume $C_l = 1$ when focusing on a single layer $l$ so that we can use $\boldsymbol{w}^{(l)}$ to denote the weight matrix for layer $l$. When dropping their superscript, $\boldsymbol{x}$ and $\boldsymbol{w}$ denote generic inputs and weights in $\mathbb{R}^K$, and $\tilde{\boldsymbol{x}}$ and $\boldsymbol{q}$ denote their quantized counterparts.

### 2.1 Post-Training Quantization

Standard quantization operators, referred to as quantizers, are commonly parameterized by zero-point $z$ and scaling factor $s$, as shown in Eq. 1 for weight tensor $\boldsymbol{w}$. Our work focuses on uniform integer quantization, where $z$ is an integer value that ensures that zero is exactly represented in the quantized domain, and $s$ is a strictly positive scalar that corresponds to the resolution (or step size) of the quantizer. Scaled values are commonly rounded to the nearest integer, denoted by $\lceil \cdot \rfloor$, and elements that exceed the representation range of the quantized domain $\mathcal{A}_b$ are clipped.

$$\mathcal{Q}(\boldsymbol{w}) := s \cdot \left( \text{clip} \left( \left\lceil \frac{\boldsymbol{w}}{s} \right\rfloor + z; \min \mathcal{A}_b, \max \mathcal{A}_b \right) - z \right) \tag{1}$$

Methods for tuning quantized models broadly fall into two paradigms: quantization-aware training (QAT) and post-training quantization (PTQ). QAT methods train or fine-tune a neural network with quantization in the loop, which often requires significant compute and sufficiently large datasets. Our work focuses on PTQ methods, which directly calibrate pre-trained models and rely on minimal data without end-to-end training. Many recent PTQ methods follow a common general structure, greedily casting and calibrating quantized models layer-by-layer or block-by-block while seeking to approximate the minimizer of the reconstruction error in Eq. 2, where $\boldsymbol{q}^*$ is the optimal set of quantized weights and $\tilde{\boldsymbol{X}}$ is the quantized counterpart of $\boldsymbol{X}$.

$$\boldsymbol{q}^* = \underset{\boldsymbol{q} \in \mathcal{A}_b^K}{\arg\min} \frac{1}{2} \|\boldsymbol{X}^T \boldsymbol{w} - \tilde{\boldsymbol{X}}^T \boldsymbol{q}\|_2^2. \tag{2}$$

Recent LLM PTQ methods often concentrate on weight-only quantization to solely minimize data storage and transfer costs (Lybrand & Saab, 2021; Frantar et al., 2022). This focus has been justified—the ever-increasing weight volume of state-of-the-art models has rendered many hyper-scale LLMs memory-bound (Zhang et al., 2022a; Biderman et al., 2023). In this context, weight-only quantization algorithms can preserve model quality and still improve end-to-end throughput just by reducing data transfer costs, even with FP16 computations (Frantar et al., 2022; Tseng et al., 2024). However, with the progression of continuous batching in cloud-based LLM serving (Yu et al., 2022) and the rise of resource-efficient sampling methods like speculative decoding (Leviathan et al., 2023), which exploit available compute when memory is the bottleneck, it is increasingly important to reduce the cost of arithmetic operations, even for hyper-scale LLMs. In these cases, weight-activation quantization presents an opportunity to not only increase throughput from

reduced data traffic, but also to benefit from accelerated computation and decreased requirements for area and power. However, as further discussed in Section 3, even weight-activation quantization may start to yield diminishing returns as narrower datatypes are used.

## 2.2 Accumulator-Aware Quantization

Let $P^*$ denote the minimum accumulator bit width required to guarantee overflow avoidance for a given dot product. Aside from universally fixing the accumulator at 32 bits (or any other arbitrary maximum width imposed by a processor), the most conservative method to calculate $P^*$ considers the width of the dot product operands. Given that inputs $\tilde{\boldsymbol{x}} \in \mathcal{A}_N^K$ and weights $\boldsymbol{q} \in \mathcal{A}_M^K$ are quantized, $P^*$ is given by Eq. 3, where $\mathbb{1}_{\text{signed}}(\tilde{\boldsymbol{x}})$ is 1 if $\tilde{\boldsymbol{x}}$ is signed and 0 otherwise.

$$P^* = \left\lceil \log_2 \left( 2^{\log_2(K) + N + M - 1 - \mathbb{1}_{\text{signed}}(\tilde{\boldsymbol{x}})} + 1 \right) + 1 \right\rceil \tag{3}$$

Note that $P^*$ increases linearly with the bit widths of the operands and logarithmically with the depth of the dot product. Thus, for a fixed neural architecture, one could heuristically manipulate the weight and activation bit widths according to Eq. 3 to reduce $P^*$. However, the quantization design space ultimately limits the minimum attainable accumulator bit width, as well as the maximum attainable accuracy for any target accumulator bit width (Colbert et al., 2023; 2024).

Colbert et al. 2024 show that one can directly target the accumulator bit width as an independent dimension of the quantization design space while still theoretically guaranteeing overflow avoidance. When accumulating $\tilde{\boldsymbol{x}}^T \boldsymbol{q}$ into a signed $P$-bit accumulator, and assuming that $\sum_i q_i = 0$, one need only constrain $\|\boldsymbol{q}\|_1$ such that:

$$\|\boldsymbol{q}\|_1 \leq \frac{2^P - 2}{2^N - 1}. \tag{4}$$

Motivated by this result, accumulator-aware QAT methods avoid overflow by constraining the $\ell_1$-norm of weights during training to ultimately restrict the range of dot product outputs during inference. Although these approaches have yielded promising results, their scope is limited to the QAT setting (Colbert et al., 2023; 2024). To the best of our knowledge, ours marks the first formal study of accumulator-aware PTQ, and the first solution to scale to modern LLMs.

# 3 Motivation

The quantization research landscape is slanted towards low-precision operands (*i.e.*, weights and activations). However, low-precision operands reduce multiplication costs significantly more than addition costs. Thus, we hypothesize that, via Amdahl's Law (Amdahl, 1967), this skewed focus will yield diminishing returns.

Indeed, recent works have already demonstrated that high-precision additions can bottleneck throughput, power, and area. For example, multiple studies have reported a 2× throughput increase when reducing the accumulator width from 32 to 16 bits on general-purpose platforms (Khudia et al., 2018b; de Bruin et al., 2020; Xie et al., 2021). Furthermore, when constraining operands to 3-bit × 1-bit multipliers, Ni et al. 2020 show that the cost of 32-bit accumulation consumes nearly 75% of the total power of their scalar MAC unit, reporting up to 3× power savings and 5× area reduction when reducing to 8-bit accumulation. Here, we further substantiate our hypothesis that reducing operand width will yield diminishing returns.

## 3.1 Reducing Operand Width Yields Diminishing Returns

Low-precision accumulation is not widely supported in modern hardware, so we substantiate our hypothesis using an accumulator-aware variant of the bit operations (BOps) cost model (Van Baalen et al., 2020; Hawks et al., 2021), presented in Eq. 5, as a hardware-agnostic proxy for power consumption. For a fixed dot product size $K$, our cost model scales quadratically with the product of the operand bit widths $M$ and $N$ but linearly with the accumulator width $P$, and increased weight sparsity $S$ only reduces the cost of additions.

$$\text{BOps} := K \times (M \times N + (1 - S) \times P) \tag{5}$$

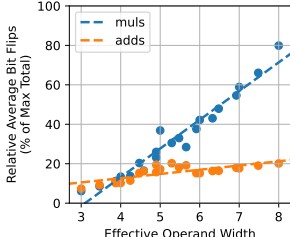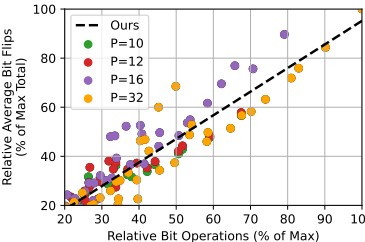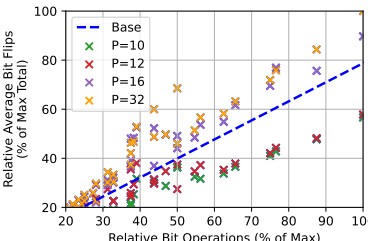

Figure 2: **Left:** Using average bit flips as a power proxy, the cost of additions (`adds`) begins to dominate that of multiplications (`muls`) as the effective operand width ($\sqrt{M \times N}$) is reduced below 4 bits in a 128-element vector MAC with a fixed 32-bit accumulator. **Center:** When varying the vector size $K$, accumulator width $P$, and operand widths $M$ and $N$, our cost model (circles) shows a strong correlation (black trendline) with average bit flips. **Right:** The baseline multiplication cost model (crosses) is unable to account for the benefits of reduced accumulator precision (blue trendline).

To the best of our knowledge, van Baalen et al. (2022) made the first connection between BOps and power by correlating multiplication costs ($K \times M \times N$) with the power consumed when executing vision models on an NPU. We extend their cost model and analysis to include the impact of accumulator-aware quantization on power consumption. To support our accumulator-aware variant, we used Yosys (Wolf, 2016) to synthesize several integer vector MAC designs while varying $K \in \{32, 64, 128\}$, $M, N \in \{3, 4, 5, 6, 7, 8\}$, and $P \in \{10, 12, 16, 32\}$. We then used the Arbolta simulator (Redding et al., 2025) to count bit flips[2] while passing discrete random Gaussian data through each synthesized design. For each $P$ and $N$, we constrain the random weights according to Eq. 4, which incidentally increases sparsity $S$ (Colbert et al., 2023; 2024).

As shown in Figure 2, our cost model exhibits a strong 96% correlation with all observed data while the baseline multiplication cost model is unable to account for the benefits of reducing accumulator width. Interestingly, we observe that addition costs begin to dominate multiplication costs when the effective operand width ($\sqrt{M \times N}$) falls below 4 bits. Moreover, our BOps cost model is consistent with the data presented by Ni et al. 2020, whereby reducing the accumulator width from 32 to 8 bits in a scalar MAC unit with a $3 \times 1$ multiplier resulted in $3\times$ power savings—our model predicts $3\times$ exactly when assuming 25% sparsity (*i.e.*, $S = 0.25$). Thus, as researchers continue to stabilize 4-bit weights and activations (Ashkboos et al., 2024; Liu et al., 2024; Zhang et al., 2025b), we suspect that neural network quantization will reach this inflection point in the near future, suggesting accumulator-aware quantization will be in the critical path for optimization as the cost of additions begins to overtake that of multiplications.

## 3.2 Limiting the Risks of Low-Precision Accumulation

Reducing the cost of additions is commonly done by reducing the accumulator bit width, which exponentially increases the risk of overflow, often introducing numerical errors that degrade model accuracy (Ni et al., 2020; Colbert et al., 2023). Existing methods that prepare quantized models for low-precision accumulation often aim to either reduce the risk of overflow (Xie et al., 2021; Li et al., 2022) or mitigate its impact on model accuracy (Ni et al., 2020). These empirical approaches rely on assumptions that limit their real-world applicability. First, empirical estimates of overflow rely on *a priori* knowledge of the input distribution, which is often impractical to assume and can even introduce vulnerabilities (Baier et al., 2019). Second, overflow behavior can vary across platforms and programs, so designing methods to mitigate the detrimental impact of one particular behavior (*e.g.*, wraparound two's complement arithmetic) limits portability. Finally, empirical approaches are unable to support applications that *require* guaranteed correctness, such as encrypted inference (Lou & Jiang, 2019), and are known to break down when overflows occur too frequently (Ni et al., 2020; Colbert et al., 2023). Thus, avoiding overflow improves reliability, portability, and model quality.

---

[2]Bit flips are known to be an effective proxy for power consumption in both compute (van Baalen et al., 2022) and memory (Bittman et al., 2018).

From the family of existing accumulator-aware QAT methods that avoid overflow, one can only apply EP-init (Colbert et al., 2024) to the PTQ setting without modification. However, EP-init has two shortcomings: (1) it relies on rounding-to-zero to ensure $|Q(w_i)| \leq |w_i|$ for all $i$, which is known to introduce catastrophic errors in PTQ (Nagel et al., 2020); and (2) it is a channel-wise projection that is not amenable to error correction, as discussed in Appendix D.2. In Section 5.1, we show that AXE better preserves model accuracy as the accumulator width is reduced, and yields a new Pareto frontier for power-efficient PTQ methods.

## 4 AXE: A General Framework for Accumulator-Aware PTQ

In the standard PTQ setting, one often assumes the quantizer parameters are fixed (*i.e.*, scaling factor $s$ and zero point $z$) and that the individual weights can move freely (Lybrand & Saab, 2021; Frantar et al., 2022). Building from these assumptions, we formalize accumulator-aware PTQ with the objective function in Eq. 6, where the optimal quantized weights $q^*$ minimize local quantization error while also satisfying an accumulator-aware $\ell_1$-norm constraint, where $Z$ is given, up to a scaling, by Eq. 4.

$$q^* = \arg\min_{q \in \mathcal{A}_b^K} \frac{1}{2} \|X^T w - \tilde{X}^T q\|_2^2 \quad \text{s.t.} \quad \|q\|_1 \leq Z \tag{6}$$

In particular, the constraint $\|q\|_1 \leq Z$ ensures, via Hölder's inequality (Hardy et al., 1952), that any inner product $|\tilde{x}^T q|$ remains appropriately bounded, as long as $\|\tilde{x}\|_\infty$ is bounded. To approximately solve this accumulator-constrained reconstruction problem, we introduce AXE—a flexible accumulator-aware quantization framework that endows overflow avoidance guarantees to the family of layerwise PTQ algorithms that greedily assign bits element-by-element (*e.g.*, GPFQ and OPTQ).

We present AXE as the following composition of functions:

$$\Phi_i := \mathcal{Q} \circ \Psi_{a_{i-1}, b_{i-1}} \circ \Pi_{\lambda^*}, \tag{7}$$

which acts on the (possibly error-corrected) weights, as shown in Algorithms 1 and 2. AXE provides accumulator-awareness by first projecting its argument onto the $\ell_1$ ball of radius $\lambda^*$ via $\Pi_{\lambda^*}$, then greedily clipping the result to the range $[a_{i-1}, b_{i-1}]$ via $\Psi_{a_{i-1}, b_{i-1}}$, and finally quantizing it to the alphabet $\mathcal{A}$, as presented in Section 4.2. Here, $\Pi$ is a per-channel, or per-tile, penalty that discourages the underlying algorithm from opportunistically selecting quantized weights with high magnitudes, and $\Psi$ is a strict per-element constraint that greedily limits the range of each selected quantized weight while error is iteratively corrected. The resulting set of quantized weights is then guaranteed to avoid overflow when accumulating its inner product with any $\tilde{X} \in \mathcal{A}_N^{K \times D}$ into $P$-bit signed registers.

In its coarsest form, AXE applies these constraints per-channel so that each dot product in the network is guaranteed to independently avoid overflow. Furthermore, without violating our constraints, we design AXE to support multi-stage accumulation in the form of tiled dot products by applying our constraints in finer granularities. Without loss of generality, we theoretically justify our solution using GPFQ, then provide accumulator-aware variants of GPFQ and OPTQ in Algorithms 1 and 2, respectively. We highlight that, to ensure $\|\tilde{x}\|_\infty$ is bounded, our accumulator-aware variants of GPFQ and OPTQ *require* quantizing activations.

### 4.1 Accumulator Constraints without Zero-Centering

Our goal with AXE is to provide a theoretical guarantee of overflow avoidance when accumulating the dot product of $q$ by any $\tilde{x} \in \mathcal{A}_N^K$ into a signed $P$-bit register. To this end, if $q$ is a zero-centered vector such that $\sum_i q_i = 0$, then it is sufficient to constrain $\|q\|_1$ to satisfy the upper bound given by Eq. 4. However, enforcing such a zero-centering constraint on a vector of integers is non-trivial in practice.

For any $\tilde{x} \in \mathcal{A}_N^K$, each element $\tilde{x}_i$ lies within the closed interval $[\mu, \nu]$ for all $i = \{1, \cdots, K\}$, and $\nu - \mu = 2^N - 1$. It follows that the maximizing vector, $u = \arg\max_{\tilde{x}} \tilde{x}^T q$, and the minimizing vector, $v = \arg\min_{\tilde{x}} \tilde{x}^T q$, are

$$u_i = \begin{cases} \nu, \text{where } q_i \geq 0 \\ \mu, \text{where } q_i < 0 \end{cases} \quad \text{and} \quad v_i = \begin{cases} \mu, \text{where } q_i \geq 0 \\ \nu, \text{where } q_i < 0 \end{cases}. \tag{8}$$

---

**Algorithm 1 Accumulator-Aware GPFQ.** Our accumulator-aware GPFQ variant quantizes $\boldsymbol{W}$ to $M$ bits given input activations $\boldsymbol{X}$ and their $N$-bit quantized counterparts $\tilde{\boldsymbol{X}}$. Note that $\boldsymbol{W}_i, \boldsymbol{V}_i \in \mathbb{R}^C$, $\boldsymbol{Q}_i \in \mathcal{A}_M^C$, $\boldsymbol{X}_i \in \mathbb{R}^D$, and $\tilde{\boldsymbol{X}}_i \in \mathcal{A}_N^D$, all interpreted as row vectors.

---

**Require:** $\boldsymbol{W} \in \mathbb{R}^{K \times C}$, $\boldsymbol{X} \in \mathbb{R}^{K \times D}$, $\tilde{\boldsymbol{X}} \in \mathcal{A}_N^{K \times D}$

| | | |
|---|---|---|
| 1: | $\boldsymbol{Q} \leftarrow 0 \in \mathcal{A}_M^{K \times C}$ | Quantized output |
| 2: | $\boldsymbol{U} \leftarrow 0 \in \mathbb{R}^{D \times C}$ | Per-sample quantization error |
| 3: | $\boldsymbol{a} \leftarrow A \in \mathbb{R}^C$, $\boldsymbol{b} \leftarrow B \in \mathbb{R}^C$ | Initialize running sums |
| 4: | $\lambda \leftarrow \text{deriveThreshold}(\boldsymbol{W})$ | Derive per-channel Lagrangian thresholds |
| 5: | **for** $i = 1, ..., K$ **do** | |
| 6: | $\quad \boldsymbol{W}_i \leftarrow \boldsymbol{W}_i \frac{\langle \tilde{\boldsymbol{X}}_i, \boldsymbol{X}_i \rangle}{\|\tilde{\boldsymbol{X}}_i\|_2^2} + \frac{\tilde{\boldsymbol{X}}_i \boldsymbol{U}}{\|\tilde{\boldsymbol{X}}_i\|_2^2}$ | Adjust for quantization error |
| 7: | $\quad \boldsymbol{V}_i \leftarrow \Psi_{\boldsymbol{a},\boldsymbol{b}} \circ \Pi_\lambda(\boldsymbol{W}_i)$ | Accumulator-aware projection & clipping |
| 8: | $\quad \boldsymbol{Q}_i \leftarrow \mathcal{Q}(\boldsymbol{V}_i)$ | Quantize weight |
| 9: | $\quad \boldsymbol{a} \leftarrow \boldsymbol{a} - \boldsymbol{Q}_i \odot \mathbb{1}_{\boldsymbol{Q}_i \geq 0}$ | Update positive range |
| 10: | $\quad \boldsymbol{b} \leftarrow \boldsymbol{b} - \boldsymbol{Q}_i \odot \mathbb{1}_{\boldsymbol{Q}_i \leq 0}$ | Update negative range |
| 11: | $\quad \boldsymbol{U} \leftarrow \boldsymbol{U} + \boldsymbol{X}_i^T \boldsymbol{W}_i - \tilde{\boldsymbol{X}}_i^T \boldsymbol{Q}_i$ | Update quantization error |
| 12: | **end for** | |
| 13: | **return** $\boldsymbol{Q}$ | |

---

Fundamentally, to avoid overflow when accumulating $\tilde{\boldsymbol{x}}^T \boldsymbol{q}$ into a $P$-bit register, the result needs to fall within the register's representation range for any $\tilde{\boldsymbol{x}} \in \mathcal{A}_N^K$. Without loss of generality, we derive our algorithm assuming a sign-magnitude accumulator for clarity and conciseness. Thus, to safely use a signed $P$-bit accumulator without overflow, both the following inequalities need to be satisfied:

$$\boldsymbol{u}^T \boldsymbol{q} \leq 2^{P-1} - 1, \quad -\boldsymbol{v}^T \boldsymbol{q} \leq 2^{P-1} - 1 \tag{9}$$

To avoid zero-centering, one could generalize the result derived by Colbert et al. 2024 such that the bound relies on a variable center, *e.g.*, $\sum_i q_i = \epsilon$. However, this precludes the use of greedy sequential algorithms where $\epsilon$ would be just as difficult to enforce as zero-centering, *i.e.*, $\epsilon = 0$. Thus, rather than constraining the center, we greedily constrain the boundaries, as further discussed in Section 4.2.

## 4.2 Accumulator-Aware GPFQ

At the $l$-th layer, GPFQ greedily selects each element $q_i$ to minimize the squared distance between the running sum $\sum_{j=1}^i q_j \tilde{\boldsymbol{X}}_j$ and its analog $\sum_{j=1}^i w_j \boldsymbol{X}_j$ such that

$$q_i^{(l)} = \underset{p \in \mathcal{A}_M}{\arg\min} \left\| \sum_{j=1}^i w_j^{(l)} \boldsymbol{X}_j^{(l)} - \sum_{j=1}^{i-1} q_j^{(l)} \tilde{\boldsymbol{X}}_j^{(l)} - p \tilde{\boldsymbol{X}}_i^{(l)} \right\|_2 \tag{10}$$

where $\tilde{\boldsymbol{X}}_i^{(l)}$ denotes samples for the $i$-th input neuron to the $l$-th layer assuming the first $l-1$ layers are quantized, and $\mathcal{A}_M$ is an $M$-bit fixed alphabet defined by the target quantization space. This simplifies to the following iteration rule, as derived by Lybrand & Saab 2021, where $u_0^{(l)} = 0$.

$$q_i^{(l)} = \mathcal{Q}\left( \frac{\langle \tilde{\boldsymbol{X}}_i^{(l)}, u_{i-1}^{(l)} + w_i^{(l)} \boldsymbol{X}_i^{(l)} \rangle}{\|\tilde{\boldsymbol{X}}_i^{(l)}\|_2^2} \right) \tag{11}$$

$$u_i^{(l)} = u_{i-1}^{(l)} + w_i^{(l)} \boldsymbol{X}_i^{(l)} - q_i^{(l)} \tilde{\boldsymbol{X}}_i^{(l)} \tag{12}$$

**Soft $\ell_1$-norm regularization penalty.** By design, greedy sequential quantization algorithms (*e.g.*, GPFQ and OPTQ) opportunistically alter weights to correct for as much error as possible in each step, often yielding high-magnitude quantized weights. However, this is unfavorable in the accumulator-aware PTQ setting as high-magnitude weights consume more of the allocated $\ell_1$-norm budget (see Eq. 4). To address this, we penalize high-magnitude weights throughout error correction via the soft $\ell_1$ penalty proposed by Zhang et al.

2023, which yields the $\ell_1$ projection given by Eq. 13, where $(\cdot)_+$ denotes the rectified linear unit (ReLU), and $\lambda > 0$ is an arbitrary tuneable regularization parameter.

$$\Pi_\lambda(\boldsymbol{w}) = \text{sign}(\boldsymbol{w})(|\boldsymbol{w}| - \lambda)_+ \tag{13}$$

Noticeably, this formulation is amenable to leverage EP-init (Colbert et al., 2024), which takes the same functional form. Thus, we determine $\lambda$ as the Lagrange multiplier derived from the optimal Euclidean projection of $\boldsymbol{w}$ onto the $\ell_1$ ball of radius $Z$, given by the following convex optimization problem, where $\boldsymbol{v}^*$ is the weight vector that minimizes the Euclidean projection of $\boldsymbol{w}$ onto the boundary of our constrained set *before* quantization.

$$\boldsymbol{v}^* = \min_{\boldsymbol{v}} \frac{1}{2} \|\boldsymbol{v} - \boldsymbol{w}\|_2^2 \quad \text{subject to} \quad \|\boldsymbol{v}\|_1 \leq Z \tag{14}$$

An efficient solution to this problem is derived by Duchi et al. (2008). Define $\rho$ as the number of non-zero elements in the optimal projection and $\boldsymbol{\mu}$ as the result of sorting the magnitudes of $\boldsymbol{w}$ in descending order, where $\mu_i = |w_j|$ for $i, j \in \{1, \cdots, K\}$. The optimal Lagrange multiplier $\lambda^*$ is

$$\lambda^* = \frac{1}{\rho} \left( \sum_{i=1}^{\rho} \mu_i - Z \right). \tag{15}$$

Thus, $\Pi_{\lambda*}(\boldsymbol{x})$ yields the optimal Euclidean projection onto our $\ell_1$ ball before quantization. As such, it is important to note that, because this projection is derived before applying error correction algorithms (*i.e.*, GPFQ or OPTQ), it cannot guarantee overflow avoidance on its own. Thus, we need our subsequent strict constraint; however, we observe our penalty consistently improves model quality (see Appendix D.2).

**Greedy $\ell_1$-norm constraint.** For clarity, and without loss of generality, we motivate our strict constraint using the case where $\tilde{\boldsymbol{x}}$ is represented with unsigned integers such that $\mu = 0$ and $\nu = 2^N - 1$. Note that this is common when following activation functions with non-negative dynamic ranges.

Let $\alpha$ denote the sum of all negative elements in $\boldsymbol{q}$, and let $\beta$ denote the sum of all positive elements in $\boldsymbol{q}$. From Eq. 9, we can similarly derive the upper bounds on $\beta$ and $-\alpha$ in the case of sign-magnitude representations. Indeed, $\boldsymbol{u}^T\boldsymbol{q} \leq 2^{P-1} - 1$ is guaranteed whenever $\beta\nu + \alpha\mu \leq 2^{P-1} - 1$, which holds in the case of unsigned activations if

$$\beta \leq \frac{2^{P-1} - 1}{2^N - 1}. \tag{16}$$

To accumulate at $P$ bits at layer $l$ without overflow, we greedily control the dot product range via

$$\Psi_{a_i^{(l)}, b_i^{(l)}}(x) = \text{clip}\left(x; a_i^{(l)}, b_i^{(l)}\right), \tag{17}$$

$$a_i^{(l)} = A^{(l)} - \alpha_i, \quad b_i^{(l)} = B^{(l)} - \beta_i, \tag{18}$$

where $\alpha_i$ denotes the sum of all negative elements in $\boldsymbol{q}$ whose index is less than $i$ and $\beta_i$ is its positive counterpart, and $A^{(l)}$ and $B^{(l)}$ (defined in Eq. 19) are respectively the upper limits of $\alpha_i$ and $\beta_i$. The range greedily enforced on $q_i$ becomes the closed interval defined by Eq 18. By independently constraining these, our accumulator-aware variant avoids overflow without explicit zero-centering. To ensure rounding errors do not compromise Eq. 16, we use

$$-A^{(l)} = B^{(l)} = \frac{2^{P-1} - 1}{2^N - 1} - \max(\Delta) \tag{19}$$

where $\max(\Delta)$ denotes the worst-case difference in magnitude caused by rounding. We note that, while our derivation considers the sign-magnitude representation for its symmetry, the separate consideration of $A^{(l)}$ and $B^{(l)}$ is useful for asymmetric representations (*e.g.*, two's complement).

At each step $i$, this yields the function composition $\Phi_i$ presented in Eq. 7, which can generally be extended to the family of layerwise adaptive rounding algorithms that includes GPFQ, OPTQ, Qronos (Zhang et al., 2025b), and others. Focusing on the former, we present the pseudo-code for our accumulator-aware variants of GPFQ and OPTQ in Algorithms 1 and 2, respectively, where we define $\Psi_{\boldsymbol{a},\boldsymbol{b}}(\boldsymbol{v})$ to denote the clipping function applied elementwise so that $(\Psi_{\boldsymbol{a},\boldsymbol{b}}(\boldsymbol{v}))_j = \Psi_{a_j,b_j}(v_j)$. Note that Algorithm 2 is in Appendix A.

### 4.3 Multi-Stage Accumulator-Aware Quantization

Our accumulator-aware constraints can be generalized to target customized datapaths beyond user-specific accumulator bit widths. To this end, we design AXE to support multi-staged accumulation as visualized in Figure 3. In such a scenario, our constraints are enforced on the quantized weights in tiles of size $T$ so that each partial dot product can be concurrently computed by an atomic MAC unit. Let $P_I$ and $P_O$ denote the inner and outer accumulator bit widths, respectively. If a $K$-dimensional dot product is executed in tiles of size $T$, where each tile is constrained to a $P_I$-bit accumulator, then the minimum $P_O$ required to guarantee overflow avoidance is

$$P_O = \lceil P_I + \log_2(K) - \log_2(T) \rceil . \tag{20}$$

The benefits of multi-stage accumulation are well-established. Khudia et al. 2018b report a $2\times$ throughput uplift on compute-bound workloads by accumulating at 16 bits in 64-element tiles instead of at 32 bits, albeit without any theoretical guarantees of overflow avoidance. Currently, inference libraries such as FBGEMM (Khudia et al., 2018a), XNNPACK (Dukahn & Barchard, 2021), and Ryzen AI (AMD, 2024) typically disable this optimization if overflows are observed too often during testing. To our knowledge, AXE provides the first mechanism to simultaneously quantize and constrain a pre-trained model for low-precision multi-staged accumulation while guaranteeing overflow avoidance, safely enabling this optimization for the first time. As shown in Section 5, this generalization is critical in maintaining the quality of billion-parameter LLMs, which often have dot products containing more than ten thousand elements.

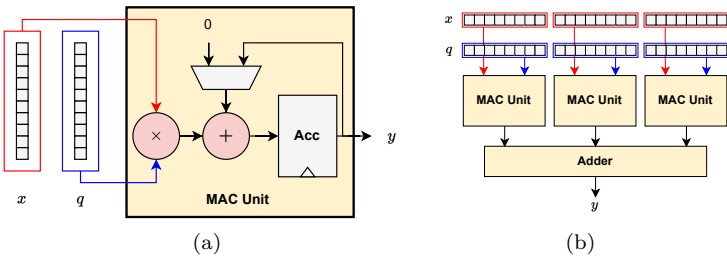

(a)             (b)

Figure 3: We visualize abstractions of (a) an atomic MAC unit and (b) parallelized multi-staged accumulation.

## 5 Experiments

The core contribution of this work is enabling a user to prepare a quantized model for low-precision accumulation in the PTQ setting via AXE. We note that the computational and memory overhead introduced by our accumulator-aware constraints is negligible in practice. Thus, our primary comparison metric is preserving model quality in challenging accumulator-aware PTQ scenarios.

**Models & Datasets.** We conduct experiments on GPT2 (Radford et al., 2019), OPT (Zhang et al., 2022a), SmolLM2 (Allal et al., 2024), Pythia (Biderman et al., 2023), and Llama3 (Dubey et al., 2024) models using WikiText2 (Merity et al., 2016) for calibration. While perplexity serves as a standard and sensitive proxy for generative quality, we complement it with an analysis of zero-shot accuracy to demonstrate generalization. When analyzing zero-shot accuracy, we use LightEval (Fourrier et al., 2023) to evaluate 4 reasoning tasks: ARC-challenge (Clark et al., 2018), HellaSwag (Zellers et al., 2019), PIQA (Bisk et al., 2020), and Winogrande (Sakaguchi et al., 2021). We note that LLM evaluation is increasingly complex, and we leave additional deeper analyses (such as impact on reasoning capabilities) for future work. Moreover, to demonstrate that our approach works well across neural architectures and tasks, we provide additional experiments with image classification models in Appendix B.

**Quantization Design Space.** We constrain our design space to uniform-precision models such that every hidden layer has the same weight, activation, and accumulator bit width, respectively denoted as $M$, $N$, and $P$. We consider 3- to 8-bit integers for both weights and activations, unlike Frantar et al. (2022) and Zhang et al. (2023), which focused on weight-only quantization. Rather than evaluating each combination of $M$

and $N$, we restrict ourselves to configurations where $N \geq M$ to reduce the cost of experimentation as such configurations tend to dominate the Pareto frontiers (Colbert et al., 2024). We implement our methods using the open source Brevitas quantization library (Franco et al., 2025), and quantize all models using a single AMD MI210 GPU[3] with 64 GB of memory. We include more implementation details in Appendix D.

## 5.1 Pareto Analysis

We first consider the scenario in which QNNs are optimized for accumulator-constrained processors in the PTQ setting. As discussed in Section 2.2, one could heuristically manipulate $M$ and $N$ according Eq. 3. To the best of our knowledge, EP-init serves as the only alternative for accumulator-aware quantization in the PTQ setting. Therefore, we use EP-init and naïve bit width manipulation as our baselines.

In Figure 4, we use Pareto frontiers to visually characterize the trade-off between accumulator bit width $P$ and WikiText2 perplexity for both GPFQ and OPTQ, respectively, across a range of models. We assume a monolithic accumulator in these experiments (i.e., $P = P_I = P_O$). For each model and each PTQ algorithm, the Pareto frontier shows the best perplexity observed for a target accumulator bit width $P$ when varying $M$ and $N$ within our design space, with the full-precision floating-point model accuracy provided for reference. Additionally, in Figure 1, we visually characterize the trade-off between accumulator bit width $P$ and relative bit operations when quantizing SmolLM2 with GPFQ. Again assuming a monolithic accumulator, the left Pareto frontier shows the lowest observed perplexity for each overflow avoidance method as we reduce $P$ while varying $M$ and $N$ within our design space with the perplexity of the full-precision floating-point model provided for reference. The right Pareto frontier similarly shows the lowest perplexity observed for a given amount of bit operations relative to W8A8 with 32-bit accumulation, which we demonstrate is a strong proxy for power consumption in Section 3. Thus, our results show that AXE establishes a Pareto-dominant frontier for both accumulator bit width and power consumption, which we further support in Appendix B using additional experiments with image classification models.

We provide a detailed breakdown of each Pareto frontier in Appendix E, where we report the perplexity of each Pareto-dominant model, their weight and activation bit widths, and resulting unstructured weight sparsity. Overall, we observe trends that are consistent with Colbert et al. (2024); the Pareto-optimal activation bit width $N$ decreases as $P$ is reduced, and sparsity conversely increases. This suggests that our accumulator-aware boundary constraints obey similar mechanics as the $\ell_1$-norm constraints of QAT methods, as our theoretical justification predicts. Moreover, as in the QAT setting, the quantization design space ultimately limits the minimum accumulator bit width attainable via naïve bit width manipulation.

Interestingly, Figure 1 shows that EP-init breaks down on SmolLM2 when weights are quantized below 5 bits, likely because EP-init relies on rounding-to-zero, which is known to introduce catastrophic quantization errors in PTQ settings (Nagel et al., 2020). We highlight that this breaking point is before the inflection point we observe in Section 3.1, where the cost of additions overtakes that of multiplications with 4-bit operands. Thus, naïve bit width manipulation dominates EP-init in power efficiency, which is consistent with our theory.

## 5.2 Scaling Analysis

The $\ell_1$-norm of an unconstrained weight vector inherently grows as its dimensionality increases. This suggests that accumulator-aware quantization scales well to strictly deeper neural architectures since the constraints tighten with width rather than depth; experimental results on the ResNet family support this hypothesis (Colbert et al., 2024). However, this also suggests that accumulator-aware quantization scales poorly in neural network families that grow in width, as is the case in transformer architectures (Zhang et al., 2022a; Biderman et al., 2023). Thus, to scale our accumulator-aware PTQ framework to billion-parameter language models, we turn to our multi-stage accumulation variant of AXE, as introduced in Section 4.3. Here, one assumes the partial sums of a dot product are concurrently computed in fixed-length tiles of size $T$. Our goal in this setting is to minimize perplexity for a target inner accumulator bit width $P_I$ that is assumed to be universal across all tiles. Hence, our accumulator width is constant even as models grow wider.

---

[3]AMD, AMD Instinct, and combinations thereof are trademarks of Advanced Micro Devices, Inc.

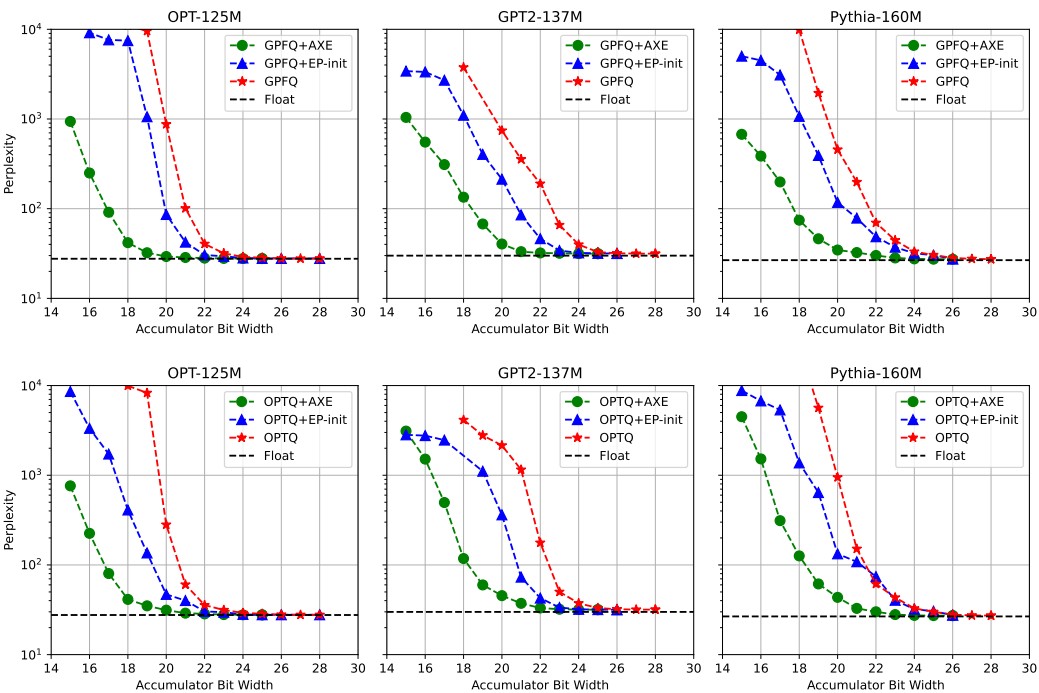

Figure 4: We show that AXE (**green circles**) yields the best trade-off between accumulator bit width and WikiText2 perplexity for several language models, namely OPT-125M, GPT2 (137M), and Pythia-160M. Note that we also show this for SmolLM2-135M in Figure 1. We compare AXE with EP-init (**blue triangles**) and naïve bit width manipulation (**red stars**) using either GPFQ (top) and OPTQ (bottom).

Rather than exploring the full quantization design space, we focus on 4-bit weights and 8-bit activations (W4A8) to maximize utility across platforms with a reasonable number of experiments as prior studies have established this configuration is generally useful (Dettmers & Zettlemoyer, 2023; Li et al., 2024). We evaluate AXE on top of both GPFQ and OPTQ using tiles of 128 elements under 16-bit accumulator constraints (note that $P_I^* = 20$ when $T = 128$ for W4A8 via Eq. 3). Prior work has also established 128 to be a generally useful tiling size: AVX-512 ISA supports $T = 32$ elements (Khudia et al., 2018a), Ryzen AI NPUs support $T = 64$ elements (AMD, 2024), and many works allocate scaling factors in groups of 128 elements (Lin et al., 2023; Liu et al., 2024). For these experiments, we apply Hadamard-based incoherence processing (Ashkboos et al., 2024; Tseng et al., 2024) to mitigate the impact of outliers when quantizing activations in LLMs.

We focus our scaling analysis on the Pythia model suite, which was specifically designed to facilitate such a study (Biderman et al., 2023). From our results in Table 1, we observe that, as model size increases, the quality of the 16-bit constrained models approaches that of the 32-bit baselines—AXE preserves 99% of the relative perplexity for Pythia-12B for both GPFQ and OPTQ, compared to 74% and 59% for Pythia-70M, respectively. We similarly observe that the gap is reduced between the 16-bit constrained models and their FP16 counterparts as model size increases—when quantizing Pythia-12B, AXE preserves 96% of the FP16 performance for both GPFQ and OPTQ, compared to the respective 54% when quantizing Pythia-70M, an impressive +42% increase in relative perplexity when scaling from 70M to 12B parameters. Under the scaling hypothesis, this suggests the narrowing accuracy gap is in part because model capacity is growing without tightening the constraints since $T$ is held constant even as $K$ increases (Pythia-12B is at most 10× wider than Pythia-70M). In Appendix D.2, we provide an ablation study targeting a monolithic 16-bit accumulator (*i.e.*, $P_O = 16$). There, we show the gap conversely increases as $K$ increases, confirming that keeping $P_I$ constant via tiled multi-stage accumulation is critical in LLM scaling.

Table 1: We report the WikiText2 perplexity results when evaluating AXE on Pythia models quantized to W4A8 for 32-bit or 16-bit accumulation in tiles of 128 elements using either GPFQ or OPTQ with Hadamard-based incoherence processing. We use 128×16b to denote $P_I = 16$ and $T = 128$, from Eq. 20.

|  |  | 70M | 160M | 410M | 1.0B | 1.4B | 2.8B | 6.9B | 12B |
|---|---|---|---|---|---|---|---|---|---|
|  | **Float16** | 41.1 | 23.7 | 14.1 | 11.7 | 10.5 | 9.2 | 8.3 | 7.7 |
| **GPFQ** | 128×32b | 56.8 | 35.2 | 19.4 | 12.3 | 11.2 | 9.5 | 8.6 | 7.9 |
|  | 128×16b | 76.4 | 55.5 | 23.2 | 12.7 | 11.8 | 9.8 | 8.7 | 8.0 |
| **OPTQ** | 128×32b | 50.6 | 32.7 | 22.4 | 12.4 | 11.4 | 9.5 | 8.5 | 7.9 |
|  | 128×16b | 85.6 | 75.5 | 34.5 | 13.1 | 12.4 | 9.9 | 8.6 | 8.0 |

Table 2: We report the WikiText2 perplexity when evaluating Llama3 models quantized to 16-bit accumulation in tiles of 128 elements with either OPTQ or GPFQ. We also demonstrate that AXE is compatible with pre-processing algorithms like SmoothQuant and Hadamard-based incoherence processing (HIP). We compare AXE (in **bold**) with a bit width manipulation baseline (Base) and provide the reference FP16 results. We use $128 \times 16b$ to denote $P_I = 16$ and $T = 128$ from Eq. 20, similarly denoting $P_I = 32$ as $128 \times 32b$.

|  | 1B | | 3B | | 8B | |
|---|---|---|---|---|---|---|
| **Float16** | 11.8 | | 9.1 | | 6.5 | |
| **128×32b** | W4A8 | AXE | W4A8 | AXE | W4A8 | AXE |
| GPFQ | 23.5 | **23.5** | 10.8 | **10.8** | 20.6 | **20.6** |
| GPFQ+SmoothQuant | 15.0 | **15.0** | 10.3 | **10.3** | 7.6 | **7.6** |
| GPFQ+HIP | 12.9 | **12.9** | 9.7 | **9.7** | 6.9 | **6.9** |
| OPTQ | 45.8 | **45.8** | 12.7 | **12.7** | 7.8 | **7.8** |
| OPTQ+SmoothQuant | 14.6 | **14.6** | 10.3 | **10.3** | 7.5 | **7.5** |
| OPTQ+HIP | 12.8 | **12.8** | 9.7 | **9.7** | 6.9 | **6.9** |
| **128×16b** | W4A4 | AXE | W4A4 | AXE | W4A4 | AXE |
| GPFQ | inf | **22.1** | inf | **14.8** | inf | **37.9** |
| GPFQ+SmoothQuant | inf | **15.4** | inf | **10.5** | inf | **7.7** |
| GPFQ+HIP | 16.1 | **13.8** | 10.8 | **10.0** | 8.0 | **7.1** |
| OPTQ | inf | **33.3** | inf | **14.1** | inf | **8.3** |
| OPTQ+SmoothQuant | inf | **15.0** | inf | **10.7** | inf | **7.6** |
| OPTQ+HIP | 15.6 | **14.0** | 10.6 | **10.0** | 7.8 | **7.1** |

### 5.3 Zero-Shot Analysis

We conclude our experiments by evaluating zero-shot reasoning on Llama3 instruction fine-tuned models, again focusing on constraining W4A8 models for 16-bit multi-stage accumulation. As discussed in Section 5.2, multi-stage accumulation is critical to scale accumulator-aware PTQ to increasingly large language models (see Appendix D.2 for ablations). Therefore, as EP-init does not support multi-stage accumulation, the only existing alternative for accumulator-aware PTQ is bit width manipulation. Note that, via Eq. 3, W4A4 guarantees overflow avoidance for 16-bit accumulation in tiles of 128 elements. Therefore, we compare AXE to W4A4 as a baseline when constraining a W4A8 model to target 16-bit accumulation in tiles of 128 elements (denoted as $128 \times 16b$); note that this corresponds to $T = 128$ and $P_I = 16$ in Eq. 20. We compare against 32-bit accumulation, which we similarly denote as $128 \times 32b$.

Since our primary comparison metric is preserving model quality in challenging accumulator-aware PTQ scenarios, we use established PTQ methods that solve orthogonal problems with the intention to create high-quality reference baselines. To this end, we demonstrate compatibility with equalization methods such as SmoothQuant (Xiao et al., 2023) and rotation-based methods such as Hadamard-based incoherence

Table 3: We report the average accuracy on zero-shot reasoning tasks when evaluating Llama3 models quantized to 16-bit accumulation in tiles of 128 elements with either OPTQ or GPFQ using Hadamard-based incoherence processing (HIP). We report the bit width manipulation baselines alongside the AXE results (in **bold**) and provide the reference FP16 results.

| | 1B | | 3B | | 8B | |
|---|---|---|---|---|---|---|
| **Float16** | 46.5 | | 53.8 | | 62.4 | |
| **128×32b** | W4A8 | AXE | W4A8 | AXE | W4A8 | AXE |
| GPFQ+HIP | 45.4 | **45.4** | 53.3 | **53.3** | 60.2 | **60.2** |
| OPTQ+HIP | 45.0 | **45.0** | 52.1 | **52.1** | 59.7 | **59.7** |
| **128×16b** | W4A4 | AXE | W4A4 | AXE | W4A4 | AXE |
| GPFQ+HIP | 41.4 | **45.1** | 50.1 | **51.2** | 57.3 | **58.6** |
| OPTQ+HIP | 42.1 | **44.5** | 49.9 | **51.1** | 56.1 | **59.3** |

processing (Ashkboos et al., 2024; Tseng et al., 2024). We provide perplexity results in Table 2 along with the FP16 reference perplexities.

AXE has the desired feature of being functionally equivalent to the base algorithm when the accumulator is large enough. As such, one should expect benefits to manifest most when targeting low-precision accumulators. Indeed, we observe that AXE improves performance in the challenging $128 \times 16b$ setting for all compositions of PTQ algorithms, with Hadamard-based incoherence processing (HIP) establishing the strongest baseline. Therefore, we use this composition of PTQ algorithms to evaluate zero-shot reasoning under accumulator constraints. We provide our results in Table 3 along with the FP16 reference accuracies.

Coupled with HIP, AXE enables low-precision accumulation for Llama3 with minimal degradation from the 32-bit baselines, preserving 98% of the relative 8B perplexity for both GPFQ and OPTQ. Furthermore, the gap between the 16-bit constrained models and their FP16 counterparts decreases as the model size increases; AXE preserves 92% of the relative perplexity for 8B compared to 86% and 84% for GPFQ and OTPQ, respectively. This result is consistent with the scaling hypothesis in Section 5.2. Finally, AXE preserves up to 95% of the FP16 zero-shot performance and 99% of the 32-bit baselines, which is up to +3% better than bit width manipulation, the only existing alternate solution that scales to billion-parameter LLMs.

# 6 Conclusions

The cost of additions overtakes that of multiplications as operand precision is reduced, as shown in Section 3. As few-bit integer representations are increasing in popularity, one expects further reducing weight and activation precision to yield diminishing returns. However, reducing the cost of additions is non-trivial due to the complexities of system design and risk of numerical errors. While prior work on accumulator-aware quantization has been limited to QAT, ours marks the first solution that extends accumulator-awareness to the PTQ setting and scales to billion-parameter LLMs. We demonstrate the flexibility of AXE by presenting and evaluating accumulator-aware variants of GPFQ and OPTQ. Furthermore, unlike prior accumulator-aware quantization methods, which assume a monolithic accumulator, we design AXE to support multi-stage accumulation for the first time. Our experiments demonstrate that AXE establishes a new state-of-the-art for accumulator-aware PTQ, yielding a Pareto-dominant frontier in the trade-off between power and accuracy.

# Statement of Broader Impact

As researchers stabilize model quality with weights and activations at 4 bits or less (Ashkboos et al., 2024; Liu et al., 2024; Ma et al., 2024; Liu et al., 2025; Zhang et al., 2025b), our work shows that adding the accumulator as a new dimension to the quantization design space can yield a new Pareto frontier that balances power consumption and model quality during inference; reducing power consumption may reduce carbon

emissions (Luccioni et al., 2023), and alleviate power and thermal bottlenecks in LLM serving (Stojkovic et al., 2025). In addition to these potential benefits, reduced accumulator precision presents a new set of practical deployment risks (see Section 3.2). Our work limits these risks by theoretically avoiding overflow and its impact on model quality. However, reducing accumulator precision, like that of weights and activations, invariably reduces the maximum attainable model quality (as evidenced by our Pareto analysis). We note that lossy compression methods (*e.g.*, quantization, pruning, distillation) can shift model behavior in subtle ways. Thus, it is important for compressed models to be further evaluated before deployment.

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

# A   Accumulator-Aware OPTQ

---

**Algorithm 2 Accumulator-Aware OPTQ.** Our accumulator-aware OPTQ variant quantizes $\boldsymbol{W}$ to $M$ bits given $\boldsymbol{H}^{-1} = \text{Cholesky}((2\tilde{\boldsymbol{X}}\tilde{\boldsymbol{X}}^T + \eta \boldsymbol{I})^{-1})$, where $\eta$ is a small dampening factor to avoid numerical issues. Following Frantar et al. 2022, we set $\eta$ to be 1% of the average diagonal value. Note that $\boldsymbol{W}_i, \boldsymbol{V}_i \in \mathbb{R}^C$ and $\boldsymbol{Q}_i \in \mathcal{A}_M^C$, all interpreted as row vectors.

---

**Require:** $\boldsymbol{W} \in \mathbb{R}^{K \times C}$, $\boldsymbol{H}^{-1} \in \mathbb{R}^{K \times K}$
1: $\boldsymbol{Q} \leftarrow 0 \in \mathcal{A}_M^{K \times C}$                                              Quantized output
2: $\boldsymbol{E} \leftarrow 0 \in \mathbb{R}^C$                             Per-channel quantization errors
3: $\boldsymbol{a} \leftarrow A \in \mathbb{R}^C$, $\boldsymbol{b} \leftarrow B \in \mathbb{R}^C$                     Initialize running sums
4: $\lambda \leftarrow \text{deriveThreshold}(\boldsymbol{W})$          Derive per-channel Lagrangian thresholds
5: **for** $i = 1, ..., K$ **do**
6:     $\boldsymbol{V}_i \leftarrow \Psi_{\boldsymbol{a},\boldsymbol{b}} \circ \Pi_\lambda(\boldsymbol{W}_i)$        Accumulator-aware projection & clipping
7:     $\boldsymbol{Q}_i \leftarrow \mathcal{Q}(\boldsymbol{V}_i)$                        Quantize processed weight
8:     $\boldsymbol{E} \leftarrow (\boldsymbol{W}_i - \boldsymbol{Q}_i)/\boldsymbol{H}_{i,i}^{-1}$            Calculate quantization error
9:     $\boldsymbol{W}_{i:K} \leftarrow \boldsymbol{W}_{i:K} - \boldsymbol{E} \cdot \boldsymbol{H}_{i,i:K}^{-1}$               Update weights
10:    $\boldsymbol{a} \leftarrow \boldsymbol{a} - \boldsymbol{Q}_i \odot \mathbb{1}_{\boldsymbol{Q}_i \geq 0}$               Update positive range
11:    $\boldsymbol{b} \leftarrow \boldsymbol{b} - \boldsymbol{Q}_i \odot \mathbb{1}_{\boldsymbol{Q}_i \leq 0}$               Update negative range
12: **end for**
13: **return** $\boldsymbol{Q}$

---

# B   Additional Results on Computer Vision Models

To further evaluate the robustness and generalizability of our approach, we include additional results on a diverse set of computer vision models—namely ResNet18 (He et al., 2016), MobileNetV2 (Sandler et al., 2018), and ViT (Dosovitskiy et al., 2020)—on the ImageNet dataset (Deng et al., 2009). We again perform an extensive Pareto analysis within the quantization design space described in Section 5.1. Once again, AXE outperforms EP-init and bit width manipulation across models using either GPFQ or OPTQ. These experiments further demonstrate that AXE maintains consistent performance across varying architectures and tasks, underscoring its applicability beyond the primary benchmarks presented in the main text. We provide details of the resulting Pareto frontiers for each model and algorithm in Tables 8 and 9.

# C   Memory-Efficient GPFQ

As discussed in Section 4.2, GPFQ approaches the standard quantization problem by traversing the neural network graph to sequentially quantize each element in each layer while iteratively correcting for quantization error. The derived iteration rule is formalized by Eqs. 11 and 12. In this standard formulation, the $i$-th quantized weight $q_i$ depends on the inner product

$$\langle \tilde{\boldsymbol{X}}_i^{(l)}, \boldsymbol{u}_{i-1}^{(l)} + w_i^{(l)} \boldsymbol{X}_i^{(l)} \rangle$$

where $\boldsymbol{X}_i^{(l)}, \tilde{\boldsymbol{X}}_i^{(l)} \in \mathbb{R}^D$ are samples for the $i$-th neuron of the inputs to layer $l$, and $\boldsymbol{u}_{i-1}^{(l)} \in \mathbb{R}^D$ is the running error from quantizing the first $i-1$ weights. Thus, at layer $l$, GPFQ requires collecting and storing $2D$ samples for the $K_l$ input neurons, and updating the running quantization error for each sample for the $C_l$ output neurons. This implies potential difficulty scaling to larger models and larger calibration sets as the memory requirements are $O(D \times (2K_l + C_l))$. Indeed, assuming 128 samples with a sequence length of 2048 at 32-bit precision, Pythia-6.9B (Biderman et al., 2023) requires a peak memory usage of roughly 30 GB at the first FFN layer excluding pre-trained weights. We set out to reduce this overhead.

We start with the observation that OPTQ is far more memory efficient. OPTQ uses the Hessian proxy $2\boldsymbol{X}\boldsymbol{X}^T$, which can be efficiently computed one sample at a time and stored as a $K_l \times K_l$ square matrix, an

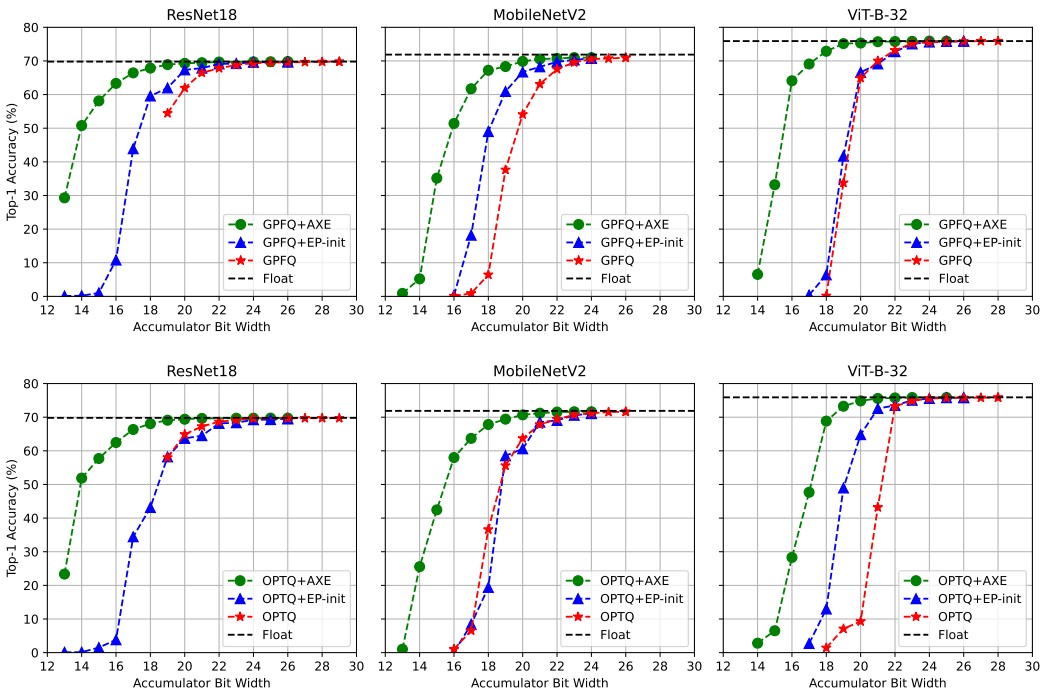

Figure 5: We show that AXE (**green circles**) yields the best trade-off between accumulator bit width and ImageNet accuracy for several image classification models, namely ResNet18, MobileNetV2, and ViT-B-32. We compare AXE with EP-init (**blue triangles**) and naïve bit width manipulation (**red stars**) using either GPFQ (top) and OPTQ (bottom).

$O(K_l \times K_l)$ memory requirement that is $36\times$ less than GPFQ for Pythia-6.9B. Thus, we reformulate GPFQ to use square matrices via mathematical manipulation of singular value decompositions.

**Theorem C.1.** *Let $\boldsymbol{H} = \left(\tilde{\boldsymbol{X}}\tilde{\boldsymbol{X}}^T\right)^{1/2}$ and $\boldsymbol{G} = \boldsymbol{X}\tilde{\boldsymbol{X}}^T$. For pre-trained weights $\boldsymbol{W} \in \mathbb{R}^{K \times C}$, quantization alphabet $\mathcal{A}$, and GPFQ function of the form of Algorithm 1, it follows that:*

$$GPFQ(\boldsymbol{W}, \boldsymbol{X}, \tilde{\boldsymbol{X}}, \mathcal{A}) = GPFQ(\boldsymbol{W}, \boldsymbol{G}\boldsymbol{H}^{-1}, \boldsymbol{H}, \mathcal{A}) \tag{21}$$

*Proof.* According to the iteration steps in Algorithm 1, it suffices to show that the argument of quantizer $\mathcal{Q}$ is unchanged after substituting $\boldsymbol{X}_i$, $\tilde{\boldsymbol{X}}_i$ with $(\boldsymbol{G}\boldsymbol{H}^{-1})_i$ and $\boldsymbol{H}_i$ respectively. Specifically, at the $i$-th iteration of $GPFQ(\boldsymbol{W}, \boldsymbol{G}\boldsymbol{H}^{-1}, \boldsymbol{H}, \mathcal{A})$, we have

$$\boldsymbol{V}_i \leftarrow \boldsymbol{W}_i \frac{\langle \boldsymbol{H}_i, (\boldsymbol{G}\boldsymbol{H}^{-1})_i \rangle}{\|\boldsymbol{H}_i\|_2^2} + \frac{\boldsymbol{H}_i \boldsymbol{U}_{i-1}}{\|\boldsymbol{H}_i\|_2^2} \tag{22}$$

where the quantization error is given by

$$\boldsymbol{U}_{i-1} = \sum_{j=1}^{i-1} (\boldsymbol{G}\boldsymbol{H}^{-1})_j^T \boldsymbol{W}_j - \boldsymbol{H}_j^T \boldsymbol{Q}_j. \tag{23}$$

Let $\boldsymbol{e}_i \in \mathbb{R}^K$ denote the vector with a 1 in the $i$-th coordinate and 0's elsewhere. It follows from $\boldsymbol{H} = \left(\tilde{\boldsymbol{X}}\tilde{\boldsymbol{X}}^T\right)^{1/2}$ and $\boldsymbol{G} = \boldsymbol{X}\tilde{\boldsymbol{X}}^T$ that

$$\|\boldsymbol{H}_i\|_2^2 = \|\boldsymbol{e}_i^T \boldsymbol{H}\|_2^2 = \boldsymbol{e}_i^T \boldsymbol{H}^2 \boldsymbol{e}_i = \boldsymbol{e}_i^T \tilde{\boldsymbol{X}}\tilde{\boldsymbol{X}}^T \boldsymbol{e}_i = \|\tilde{\boldsymbol{X}}_i\|_2^2,$$

$$\boldsymbol{H}_i(\boldsymbol{G}\boldsymbol{H}^{-1})_j^T = \boldsymbol{e}_i^T \boldsymbol{H}(\boldsymbol{e}_j^T \boldsymbol{G}\boldsymbol{H}^{-1})^T = \boldsymbol{e}_i^T \boldsymbol{G}^T \boldsymbol{e}_j = \boldsymbol{e}_i^T \tilde{\boldsymbol{X}}\boldsymbol{X}^T \boldsymbol{e}_j = \tilde{\boldsymbol{X}}_i \boldsymbol{X}_j^T,$$

and

$$\boldsymbol{H}_i \boldsymbol{H}_j^T = \boldsymbol{e}_i^T \boldsymbol{H} (\boldsymbol{e}_j^T \boldsymbol{H})^T = \boldsymbol{e}_i^T \boldsymbol{H}^2 \boldsymbol{e}_j = \boldsymbol{e}_i^T \tilde{\boldsymbol{X}} \tilde{\boldsymbol{X}}^T \boldsymbol{e}_j = \tilde{\boldsymbol{X}}_i \tilde{\boldsymbol{X}}_j^T.$$

Plugging above identities into equation 22 and equation 23, we obtain

$$\boldsymbol{V}_i \leftarrow \boldsymbol{W}_i \frac{\langle \tilde{\boldsymbol{X}}_i, \boldsymbol{X}_i \rangle}{\|\tilde{\boldsymbol{X}}_i\|_2^2} + \frac{\tilde{\boldsymbol{X}}_i \hat{\boldsymbol{U}}_{i-1}}{\|\tilde{\boldsymbol{X}}_i\|_2^2} \tag{24}$$

with $\hat{\boldsymbol{U}}_{i-1} = \sum_{j=1}^{i-1} \boldsymbol{X}_j^T \boldsymbol{W}_j - \tilde{\boldsymbol{X}}_j^T \boldsymbol{Q}_j$. Since $\boldsymbol{V}_i$ in equation 24 is identical with the $i$-th quantization argument in GPFQ$(\boldsymbol{W}, \boldsymbol{X}, \tilde{\boldsymbol{X}}, \mathcal{A})$, both algorithms derive the same quantized weights $\boldsymbol{Q}_i = \mathcal{Q}(\boldsymbol{V}_i)$. $\qquad\square$

At layer $l$, this memory-efficient GPFQ formulation requires collecting and storing $G$, $H$, and $U$, which are each $K_l \times K_l$ matrices, reducing to an $O(K_l \times K_l)$ memory requirement that is $12\times$ less than the standard GPFQ formulation for Pythia-6.9B. We leverage this functionally equivalent formulation for our LLM evaluations in Section 5.2.

## D  Experimental Details & Ablations

### D.1  Hyperparameters & Quantization Schemes

Below, we provide a detailed description of the quantization schemes and the specific hyperparameters used in our experiments. As discussed in Section 5, we consider pre-trained autoregressive language generation models that are respectively made publicly available via the HuggingFace (Wolf et al., 2020) libraries. All models are quantized via the Brevitas (Franco et al., 2025) quantization library using a single AMD MI210 GPU with 64 GB of memory.

We leverage the unmodified implementations of the various LLMs discussed in Section 5 as provided by HuggingFace (Wolf et al., 2020), as well as their pre-trained floating-point checkpoints and datasets (Lhoest et al., 2021). We use drop-in replacements for all linear layers in the networks except the embedding layer or final prediction head, leaving them at full-precision floating-point. As is common practice (Frantar et al., 2022), we build our calibration set using 128 samples randomly selected from the WikiText2 dataset (Merity et al., 2016) without replacement using a fixed sequence length of 2048 tokens for all models except GPT2 (Radford et al., 2019), which is restricted to a maximum sequence length of 1024 by the library.

**Implementation Details.** When quantizing weights with OPTQ or GPFQ, we do so in descending order according to the diagonal value of the Hessian proxy ($2\boldsymbol{X}\boldsymbol{X}^T$ by our notation in Section 2) (IST-DASLab, 2022; Lin et al., 2023; Chee et al., 2024). For GPFQ, we find that the peak memory utilization of the algorithm in its standard form ultimately limits its evaluation on billion-parameter LLMs. Thus, we introduce a functionality equivalent memory-efficient reformulation to enable the algorithm to scale to larger models (see Appendix C), which we use in our experiments. When inverting $H$ in both OPTQ and GPFQ, we use the standard dampening factor of 1% of the average of its diagonal. We use the AXE variants of GPFQ and OPTQ introduced in Section 4. When evaluating EP-init, we do so after applying the baseline OPTQ or GPFQ algorithms.

**Quantization Scheme.** We quantize activations asymmetrically, tuning $z$ to the lowest 99-th percentile based on the calibration data. While AXE is not reliant on symmetric weight quantization, we eliminate zero-points in all weight quantizers such that $z = 0$, as is common practice so as to avoid computational overhead of cross-terms (Nagel et al., 2021; Zhang et al., 2022b). Throughout our experiments, we adopt full-precision floating-point scaling factors defined as $s = \max(\boldsymbol{w})/(2^{b-1} - 1)$, where $\max(\boldsymbol{w})$ is calculated *per-channel* for the weights and *per-token* for the activations quantized for $b$-bit quantization.

To quantize our models, we first load the pre-trained checkpoint and merge normalization layers when possible. When applying SmoothQuant (Xiao et al., 2023) or Hadamard-based incoherence processing (Ashkboos et al., 2024), we do so before calibrating the scaling factors and zero-points. When applying SmoothQuant, we perform a light grid search over its $\alpha$ parameter and find $\alpha = 0.4$ to generally perform the best on average for Llama3, so we use this for all models. We then apply either GPFQ or OPTQ (with or without AXE).

## D.2 Ablation Studies

**Impact of error correction and choice of rounding function.** Previous reports had suspected EP-init is limited by its reliance on the round-to-zero (RTZ) rounding function (Colbert et al., 2023; 2024), which has been shown to be a poor choice (Nagel et al., 2020). AXE removes this reliance and also enables greedy error correction. We design an ablation study to isolate the impact of RTZ and error correction. We quantize OPT-125M (Zhang et al., 2022a) and Pythia-160M (Biderman et al., 2023) to 4-bit weights and 8-bit activations while targeting 20-bit accumulation since our Pareto front shows this configuration to be both reasonable and challenging. We evaluate AXE with round-to-zero (AXE-RTZ) and AXE with round-to-nearest (AXE-RTN). We report the results in Table 4. We interpret the gap between EP-init and AXE-RTZ as the benefit of error correction, and the gap between AXE-RTZ and AXE-RTN as the benefit of the selected rounding function. We observe that error correction has a greater impact than rounding function selection for GPFQ, but we observe the opposite for OPTQ. Finally, we evaluate AXE with our hard constraint only (AXE-HCO), that is $\Psi_{a_{i-1}, b_{i-1}}$ from Eq. 17, to isolate the impact of our soft constraint, which is not necessary for guaranteeing overflow avoidance. We interpret the gap between AXE-RTN and AXE-HCO as the impact of our soft constraint, which consistently provides improved or maintained performance.

**Multi-stage vs. monolithic accumulation.** In Section 5.2, we analyze how our accumulator constraints scale to increasingly large language models within the Pythia model suite (Biderman et al., 2023). There, we discuss our observation that, as model size increases, the quality of the accumulator-constrained models approaches that of the unconstrained baselines for both GPFQ and OPTQ. This suggests the narrowing gap in perplexity is in part because model capacity is growing without tightening the constraints. To verify this, we perform an ablation study targeting a monolithic 16-bit accumulator (*i.e.*, $P_I = P_O = 16$). We quantize all Pythia models up to Pythia-1B using either OPTQ or GPFQ, and report the results in Table 5. Not only do we observe significant instability, we also observe a $7.4\times$ regression in perplexity between Pythia-70M and Pythia-1B, confirming that fixing $P_I$ improves scaling as models grow wider.

Table 4: We evaluate round-to-nearest (RTN) and round-to-zero (RTZ) to directly compare against EP-init. We also evaluate AXE with our hard constraint only (HCO) to isolate the impact of our soft constraint. All models are quantized to W4A8 while targeting a 20-bit monolitic accumulator (*i.e.*, $P_O = 20$).

| Algorithm | Model | EP-init | AXE-RTZ | AXE-RTN | AXE-HCO |
|---|---|---|---|---|---|
| GPFQ | **OPT-125M** | 8828.3 | 165.2 | 31.9 | 31.9 |
| | **Pythia-160M** | 2500.2 | 211.0 | 43.0 | 49.2 |
| OPTQ | **OPT-125M** | 998.6 | 539.3 | 37.1 | 70.0 |
| | **Pythia-160M** | 4524.4 | 1798.7 | 84.9 | 194.8 |

Table 5: We evaluate AXE using Pythia models quantized to W4A8 when targeting a monolithic 16-bit accumulator (*i.e.*, $P_O = 16$). Note that this is in direct contrast with Table 1, which targets multi-stage accumulation (*i.e.*, $P_I = 16$).

| Algorithm | 70M | 160M | 410M | 1B |
|---|---|---|---|---|
| **GPFQ** | 4397 | 7135 | 10496 | 32601 |
| **OPTQ** | 2438 | 4439 | 9759 | 34387 |

# E Pareto Frontier Details

We provide the detailed Pareto frontiers visualized in Figures 4 and 5 for GPFQ and OPTQ. For each model, we report the perplexity, quantization configuration, and unstructured weight sparsity.

Table 6: **GPFQ:** We provide the test perplexity (PPL) and quantization configuration of the Pareto-optimal models that form the frontiers visualized in Figure 4. Note that $M$ and $N$ respectively denote the weight and activation bit widths.

| Model | P | GPFQ | | | GPFQ+EP-init | | | GPFQ+AXE | | |
|---|---|---|---|---|---|---|---|---|---|---|
| | | **PPL** | $(M,N)$ | **Sparsity** | **PPL** | $(M,N)$ | **Sparsity** | **PPL** | $(M,N)$ | **Sparsity** |
| **SmolLM2-135M** (Float: 14.4) | 16 | - | - | - | 13152.0 | (3,4) | 71.7 | **79.8** | **(4,5)** | **24.3** |
| | 17 | 57568.0 | (3,3) | 65.9 | 10816.0 | (3,4) | 71.7 | **27.8** | **(4,6)** | **22.7** |
| | 18 | 1075.0 | (3,4) | 48.8 | 283.7 | (4,5) | 38.5 | **22.5** | **(5,6)** | **13.2** |
| | 19 | 171.5 | (3,5) | 40.3 | 137.8 | (4,6) | 35.5 | **19.0** | **(5,6)** | **10.1** |
| | 20 | 61.9 | (3,6) | 38.0 | 126.4 | (6,6) | 13.0 | **16.0** | **(5,7)** | **9.9** |
| | 21 | 23.4 | (4,6) | 19.9 | 19.8 | (6,6) | 9.9 | **14.8** | **(6,7)** | **5.0** |
| | 22 | 19.0 | (5,6) | 10.1 | 16.3 | (6,7) | 9.7 | **14.2** | **(6,8)** | **4.9** |
| | 23 | 16.0 | (5,7) | 9.9 | 15.1 | (7,7) | 4.9 | **14.0** | **(7,8)** | **2.6** |
| | 24 | 14.8 | (6,7) | 5.0 | 14.4 | (7,8) | 4.9 | **13.9** | **(8,8)** | **1.2** |
| | 32 | 14.0 | (8,8) | 1.2 | 14.1 | (8,8) | 2.4 | **13.9** | **(8,8)** | **1.2** |
| **OPT-125M** (Float: 27.7) | 16 | - | - | - | 9148.8 | (3,4) | 76.5 | **249.8** | **(3,6)** | **55.6** |
| | 17 | - | - | - | 7624.6 | (3,4) | 72.7 | **91.2** | **(4,6)** | **37.9** |
| | 18 | 11007.2 | (3,3) | 58.3 | 7471.2 | (3,5) | 75.5 | **41.8** | **(4,6)** | **27.8** |
| | 19 | 9567.6 | (3,4) | 54.5 | 1059.3 | (5,6) | 39.1 | **32.3** | **(4,7)** | **27.0** |
| | 20 | 874.4 | (3,5) | 50.5 | 86.1 | (5,6) | 29.8 | **29.3** | **(5,7)** | **15.7** |
| | 21 | 101.0 | (3,6) | 46.4 | 42.4 | (5,7) | 28.1 | **28.6** | **(5,8)** | **15.6** |
| | 22 | 40.5 | (4,6) | 26.3 | 30.4 | (6,7) | 16.0 | **28.1** | **(6,8)** | **9.6** |
| | 23 | 31.8 | (4,7) | 25.9 | 29.5 | (6,8) | 15.9 | **27.9** | **(6,8)** | **8.6** |
| | 24 | 29.0 | (5,7) | 14.7 | 28.2 | (7,8) | 9.5 | **27.8** | **(7,8)** | **5.4** |
| | 32 | **27.8** | **(8,8)** | **3.8** | **27.8** | **(8,8)** | **5.3** | **27.8** | **(8,8)** | **3.8** |
| **GPT2-137M** (Float: 29.9) | 16 | - | - | - | 3345.8 | (3,3) | 93.2 | **552.4** | **(3,6)** | **55.4** |
| | 17 | - | - | - | 2705.3 | (3,6) | 75.1 | **310.1** | **(3,7)** | **52.8** |
| | 18 | 3760.3 | (3,3) | 82.3 | 1100.5 | (4,5) | 52.9 | **134.3** | **(4,7)** | **34.9** |
| | 19 | 2782.2 | (3,4) | 43.9 | 402.9 | (4,6) | 47.3 | **67.5** | **(4,7)** | **25.6** |
| | 20 | 742.4 | (3,5) | 55.3 | 213.2 | (4,7) | 44.3 | **40.4** | **(4,8)** | **24.5** |
| | 21 | 356.2 | (3,6) | 48.8 | 85.2 | (5,7) | 24.9 | **33.2** | **(5,8)** | **13.2** |
| | 22 | 189.9 | (4,6) | 26.4 | 46.3 | (5,8) | 23.8 | **32.1** | **(6,8)** | **7.3** |
| | 23 | 65.8 | (4,7) | 24.7 | 34.2 | (6,8) | 13.0 | **31.8** | **(6,8)** | **6.3** |
| | 24 | 39.8 | (4,8) | 23.8 | 32.1 | (7,8) | 7.1 | **31.5** | **(7,8)** | **3.2** |
| | 32 | **31.5** | **(8,8)** | **1.6** | 31.6 | (8,8) | 3.2 | **31.5** | **(8,8)** | **1.6** |
| **Pythia-160M** (Float: 26.7) | 16 | - | - | - | 4501.1 | (3,4) | 76.8 | **386.0** | **(3,6)** | **53.2** |
| | 17 | - | - | - | 3095.1 | (3,5) | 72.5 | **198.6** | **(3,6)** | **46.3** |
| | 18 | 9887.1 | (3,3) | 49.4 | 1070.2 | (4,5) | 46.7 | **74.5** | **(4,6)** | **25.1** |
| | 19 | 1946.8 | (3,4) | 49.8 | 391.7 | (4,6) | 42.9 | **46.2** | **(4,7)** | **24.4** |
| | 20 | 456.2 | (3,5) | 47.8 | 117.5 | (5,6) | 23.6 | **34.6** | **(5,7)** | **13.3** |
| | 21 | 198.3 | (3,6) | 45.1 | 78.5 | (5,7) | 23.4 | **32.4** | **(5,8)** | **13.3** |
| | 22 | 69.6 | (4,6) | 23.5 | 48.6 | (5,7) | 21.2 | **30.1** | **(6,8)** | **7.8** |
| | 23 | 44.4 | (4,7) | 22.6 | 37.2 | (6,8) | 13.0 | **28.2** | **(6,8)** | **5.5** |
| | 24 | 33.2 | (5,7) | 11.3 | 31.8 | (7,8) | 7.4 | **27.6** | **(7,8)** | **2.8** |
| | 32 | **27.4** | **(8,8)** | **1.4** | 27.5 | (8,8) | 2.7 | **27.4** | **(8,8)** | **1.4** |

Table 7: **OPTQ:** We provide the test perplexity (PPL) and quantization configuration of the Pareto-optimal models that form the frontiers visualized in Figure 4. Note that $M$ and $N$ respectively denote the weight and activation bit widths.

| Model | P | OPTQ | | | OPTQ+EP-init | | | OPTQ+AXE | | |
|---|---|---|---|---|---|---|---|---|---|---|
| | | **PPL** | $(M,N)$ | **Sparsity** | **PPL** | $(M,N)$ | **Sparsity** | **PPL** | $(M,N)$ | **Sparsity** |
| **OPT-125M** (Float: 27.7) | 16 | - | - | - | 3333.8 | (4,5) | 62.2 | **225.0** | **(3,6)** | **52.8** |
| | 17 | - | - | - | 1722.6 | (4,5) | 53.6 | **80.2** | **(3,6)** | **45.7** |
| | 18 | 9942.5 | (3,3) | 54.5 | 409.8 | (5,5) | 36.1 | **41.3** | **(4,6)** | **26.6** |
| | 19 | 8278.3 | (3,4) | 47.5 | 136.0 | (5,6) | 35,7 | **35.0** | **(5,6)** | **15.1** |
| | 20 | 281.1 | (3,5) | 45.5 | 46.9 | (5,6) | 26.8 | **31.3** | **(5,6)** | **14.2** |
| | 21 | 60.4 | (3,6) | 44.7 | 40.1 | (5,7) | 26.8 | **29.0** | **(5,7)** | **14.2** |
| | 22 | 35.7 | (4,6) | 25.8 | 30.3 | (6,7) | 15.6 | **28.5** | **(5,8)** | **14.2** |
| | 23 | 31.5 | (5,6) | 14.6 | 29.7 | (6,8) | 15.6 | **28.0** | **(6,8)** | **8.6** |
| | 24 | 29.2 | (5,7) | 14.6 | 28.1 | (7,8) | 9.5 | **27.8** | **(7,8)** | **5.4** |
| | 32 | **27.8** | **(8,8)** | **2.2** | **27.8** | **(8,8)** | **5.6** | **27.8** | **(8,8)** | **2.2** |
| **GPT2-137M** (Float: 29.9) | 16 | - | - | - | 2765.6 | (4,4) | 52.6 | **1513.6** | **(4,5)** | **34.0** |
| | 17 | - | - | - | 2465.0 | (4,4) | 49.0 | **496.4** | **(3,6)** | **43.4** |
| | 18 | 4140.7 | (3,3) | 59.3 | 2465.0 | (4,4) | 49.0 | **117.9** | **(4,6)** | **24.2** |
| | 19 | 2782.2 | (3,4) | 43.9 | 1108.4 | (5,6) | 34.5 | **59.9** | **(4,7)** | **24.2** |
| | 20 | 2149.8 | (4,4) | 26.0 | 361.7 | (4,7) | 43.6 | **45.5** | **(5,7)** | **13.1** |
| | 21 | 1153.8 | (4,5) | 24.7 | 73.1 | (5,7) | 24.7 | **37.3** | **(5,8)** | **13.2** |
| | 22 | 176.9 | (4,6) | 24.0 | 42.7 | (5,8) | 24.5 | **33.1** | **(6,8)** | **12.2** |
| | 23 | 50.1 | (4,7) | 23.2 | 33.5 | (6,8) | 13.4 | **32.1** | **(6,8)** | **6.2** |
| | 24 | 37.4 | (5,7) | 12.2 | 32.0 | (7,8) | 7.3 | **31.8** | **(7,8)** | **3.1** |
| | 32 | 31.8 | (8,8) | 1.6 | **31.7** | **(8,8)** | **3.3** | **31.7** | **(8,8)** | **1.6** |
| **Pythia-160M** (Float: 26.7) | 16 | - | - | - | 6739.6 | (4,6) | 79.7 | **1521.2** | **(3,5)** | **41.7** |
| | 17 | - | - | - | 5345.7 | (4,5) | 49.9 | **311.7** | **(4,5)** | **22.9** |
| | 18 | 27098.1 | (3,3) | 40.5 | 1372.4 | (4,5) | 41.1 | **126.1** | **(4,6)** | **23.1** |
| | 19 | 5644.0 | (3,4) | 40.3 | 641.2 | (4,6) | 41.0 | **61.4** | **(4,6)** | **21.3** |
| | 20 | 948.4 | (3,5) | 40.1 | 132.9 | (5,6) | 23.4 | **43.5** | **(5,6)** | **10.9** |
| | 21 | 151.3 | (4,5) | 21.4 | 108.5 | (5,7) | 23.5 | **32.8** | **(5,7)** | **10.9** |
| | 22 | 61.4 | (4,6) | 21.3 | 74.1 | (5,7) | 22.0 | **30.0** | **(5,8)** | **10.9** |
| | 23 | 43.3 | (5,6) | 10.9 | 40.4 | (6,8) | 13.0 | **28.0** | **(6,8)** | **5.5** |
| | 24 | 32.8 | (5,7) | 10.9 | 32.1 | (7,8) | 7.5 | **27.4** | **(7,8)** | **2.7** |
| | 32 | **27.2** | **(8,8)** | **1.4** | 27.6 | (8,8) | 2.9 | **27.2** | **(8,8)** | **1.4** |

Table 8: **GPFQ:** We provide the top-1 test accuracy and quantization configuration of the Pareto-optimal models that form the frontiers visualized in Figure 5. Note that $M$ and $N$ respectively denote the weight and activation bit widths.

| Model | P | GPFQ | | | GPFQ+EP-init | | | GPFQ+AXE | | |
|---|---|---|---|---|---|---|---|---|---|---|
| | | Top-1 | $(M,N)$ | Sparsity | Top-1 | $(M,N)$ | Sparsity | Top-1 | $(M,N)$ | Sparsity |
| **ResNet18** (Float: 69.8) | 14 | - | - | - | 0.2 | (3,3) | 83.5 | **50.8** | **(3,3)** | **56.9** |
| | 15 | - | - | - | 1.0 | (3,3) | 74.3 | **58.1** | **(3,4)** | **58.6** |
| | 16 | - | - | - | 10.8 | (3,3) | 62.3 | **63.3** | **(4,4)** | **40.4** |
| | 17 | - | - | - | 43.9 | (4,4) | 57.2 | **66.5** | **(4,4)** | **29.9** |
| | 18 | - | - | - | 59.6 | (4,4) | 43.5 | **67.9** | **(4,5)** | **30.7** |
| | 19 | 54.5 | (3,3) | 47.4 | 62.0 | (4,5) | 44.1 | **68.9** | **(5,5)** | **16.7** |
| | 20 | 62.0 | (3,4) | 49.5 | 67.3 | (5,5) | 27.6 | **69.3** | **(5,6)** | **16.8** |
| | 21 | 66.5 | (4,4) | 29.8 | 67.9 | (5,6) | 27.8 | **69.5** | **(6,6)** | **8.7** |
| | 22 | 67.9 | (4,5) | 30.5 | 69.1 | (6,6) | 15.7 | **69.7** | **(6,7)** | **8.7** |
| | 32 | **69.8** | **(8,8)** | **2.3** | 69.7 | (8,8) | 4.4 | **69.8** | **(8,8)** | **2.3** |
| **MobileNetV2** (Float: 71.9) | 14 | - | - | - | - | - | - | **5.2** | **(4,4)** | **17.6** |
| | 15 | - | - | - | - | - | - | **35.1** | **(4,5)** | **16.7** |
| | 16 | 0.1 | (3,3) | 33.8 | 0.3 | (4,4) | 26.5 | **51.4** | **(4,6)** | **16.6** |
| | 17 | 0.9 | (3,4) | 30.2 | 18.2 | (5,5) | 16.7 | **61.7** | **(5,6)** | **9.6** |
| | 18 | 6.4 | (3,5) | 28.8 | 49.0 | (5,6) | 16.8 | **67.2** | **(5,7)** | **9.7** |
| | 19 | 37.6 | (4,5) | 15.2 | 60.9 | (6,6) | 10.1 | **68.3** | **(5,7)** | **7.9** |
| | 20 | 54.1 | (4,6) | 14.9 | 66.7 | (6,7) | 10.3 | **69.9** | **(6,8)** | **5.9** |
| | 21 | 64.3 | (5,6) | 8.0 | 68.2 | (7,7) | 6.5 | **70.6** | **(7,8)** | **4.2** |
| | 22 | 68.3 | (5,7) | 7.9 | 69.7 | (7,8) | 6.4 | **70.7** | **(7,8)** | **2.3** |
| | 32 | **71.0** | **(8,8)** | **1.4** | 70.8 | (8,8) | 2.7 | **71.0** | **(8,8)** | **1.4** |
| **ViT-B-32** (Float: 75.9) | 16 | - | - | - | 0.1 | (3,3) | 89.0 | **64.1** | **(3,5)** | **46.9** |
| | 17 | - | - | - | 0.4 | (3,4) | 67.5 | **69.1** | **(4,5)** | **31.3** |
| | 18 | 0.2 | (3,3) | 51.2 | 6.4 | (4,5) | 47.5 | **72.9** | **(3,7)** | **46.0** |
| | 19 | 35.7 | (3,4) | 47.1 | 41.7 | (4,5) | 43.4 | **75.1** | **(4,7)** | **30.0** |
| | 20 | 64.9 | (3,5) | 45.0 | 66.6 | (5,5) | 27.8 | **75.3** | **(4,8)** | **29.2** |
| | 21 | 70.0 | (4,5) | 28.7 | 69.1 | (4,7) | 40.2 | **75.7** | **(5,8)** | **18.8** |
| | 22 | 73.2 | (3,7) | 43.6 | 72.7 | (5,8) | 27.3 | **75.8** | **(6,8)** | **12.3** |
| | 23 | 75.2 | (4,7) | 27.4 | 75.1 | (6,8) | 17.2 | **75.8** | **(6,8)** | **9.3** |
| | 24 | 75.6 | (5,7) | 16.5 | 75.6 | (6,8) | 15.1 | **75.9** | **(8,8)** | **6.3** |
| | 32 | **75.9** | **(8,8)** | **5.4** | 75.9 | (8,8) | 5.2 | **75.9** | **(8,8)** | **5.4** |

Table 9: **OPTQ:** We provide the top-1 test accuracy and quantization configuration of the Pareto-optimal models that form the frontiers visualized in Figure 5. Note that $M$ and $N$ respectively denote the weight and activation bit widths.

| Model | P | OPTQ | | | OPTQ+EP-init | | | OPTQ+AXE | | |
|---|---|---|---|---|---|---|---|---|---|---|
| | | Top-1 | $(M,N)$ | Sparsity | Top-1 | $(M,N)$ | Sparsity | Top-1 | $(M,N)$ | Sparsity |
| **ResNet18** (Float: 69.8) | 14 | - | - | - | 0.1 | (3,3) | 82.6 | **51.9** | **(3,3)** | **58.9** |
| | 15 | - | - | - | 1.5 | (3,3) | 71.6 | **57.7** | **(3,4)** | **59.9** |
| | 16 | - | - | - | 3.8 | (3,4) | 72.0 | **62.5** | **(4,4)** | **39.1** |
| | 17 | - | - | - | 34.4 | (4,4) | 55.6 | **66.4** | **(4,4)** | **32.0** |
| | 18 | - | - | - | 43.1 | (4,4) | 49.9 | **68.1** | **(4,5)** | **32.0** |
| | 19 | 58.0 | (3,3) | 54.1 | 58.2 | (5,5) | 37.7 | **69.1** | **(5,5)** | **17.1** |
| | 20 | 64.9 | (3,4) | 54.1 | 63.7 | (5,5) | 30.3 | **69.4** | **(5,6)** | **17.0** |
| | 21 | 67.2 | (4,4) | 31.9 | 64.5 | (6,6) | 25.0 | **69.5** | **(6,6)** | **8.7** |
| | 22 | 68.6 | (4,5) | 31.9 | 68.2 | (6,6) | 16.9 | **69.6** | **(6,7)** | **8.7** |
| | 32 | **69.7** | **(8,8)** | **2.3** | 69.7 | (8,8) | 4.4 | **69.7** | **(8,8)** | **2.3** |
| **MobileNetV2** (Float: 71.9) | 14 | - | - | - | - | - | - | **25.2** | **(4,4)** | **16.2** |
| | 15 | - | - | - | - | - | - | **42.5** | **(4,5)** | **16.4** |
| | 16 | 1.0 | (3,3) | 26.6 | 0.1 | (4,4) | 32.4 | **58.0** | **(5,5)** | **9.5** |
| | 17 | 6.6 | (3,4) | 26.5 | 8.4 | (5,5) | 17.0 | **63.7** | **(5,5)** | **8.0** |
| | 18 | 36.6 | (4,4) | 28.8 | 19.4 | (5,6) | 17.2 | **67.8** | **(5,6)** | **7.9** |
| | 19 | 55.6 | (4,5) | 15.2 | 58.5 | (6,6) | 10.2 | **69.4** | **(5,7)** | **8.0** |
| | 20 | 63.7 | (5,5) | 14.9 | 60.6 | (6,7) | 10.2 | **70.7** | **(6,7)** | **4.2** |
| | 21 | 67.8 | (5,6) | 7.9 | 68.5 | (7,7) | 6.5 | **71.2** | **(7,7)** | **2.4** |
| | 22 | 69.4 | (5,7) | 7.9 | 69.0 | (7,8) | 6.5 | **71.5** | **(7,8)** | **2.4** |
| | 32 | **71.6** | **(8,8)** | **1.4** | 71.1 | (8,8) | 2.9 | **71.6** | **(8,8)** | **1.4** |
| **ViT-B-32** (Float: 75.9) | 16 | - | - | - | 0.1 | (3,3) | 89.0 | **28.3** | **(3,6)** | **63.2** |
| | 17 | - | - | - | 2.7 | (3,5) | 66.8 | **47.4** | **(3,6)** | **53.0** |
| | 18 | 1.4 | (3,3) | 52.5 | 13.0 | (3,6) | 65.4 | **68.8** | **(3,7)** | **53.1** |
| | 19 | 7.1 | (3,4) | 50.8 | 48.9 | (4,6) | 48.1 | **73.3** | **(3,7)** | **48.2** |
| | 20 | 10.2 | (3,5) | 54.1 | 64.8 | (4,7) | 47.3 | **74.8** | **(4,7)** | **32.8** |
| | 21 | 47.7 | (3,6) | 53.0 | 72.6 | (5,7) | 33.6 | **75.6** | **(4,8)** | **32.4** |
| | 22 | 73.3 | (3,7) | 48.2 | 73.5 | (5,8) | 33.2 | **75.7** | **(6,8)** | **22.0** |
| | 23 | 74.8 | (4,7) | 32.8 | 75.0 | (6,8) | 23.2 | **75.9** | **(6,8)** | **14.4** |
| | 24 | 75.6 | (4,8) | 32.4 | 75.6 | (6,8) | 21.5 | **75.9** | **(8,8)** | **9.4** |
| | 32 | **75.9** | **(8,8)** | **8.6** | 75.8 | (8,8) | 10.0 | **75.9** | **(8,8)** | **8.6** |

