# OpenReview forum: "Accumulator-Aware Post-Training Quantization for Large Language Models"
_TMLR — Accepted by TMLR_

### Review · Reviewer_vzxV · 2025-09-16

**Summary Of Contributions:**

The paper introduces AXE, the first accumulator-aware post-training quantization (PTQ) framework that guarantees overflow avoidance.

First Formalization of Accumulator-Aware PTQ: The paper provides the first formal study and definition of accumulator-aware PTQ, extending the concept beyond the previously limited quantization-aware training (QAT) setting.

Introduction of AXE for Overflow Avoidance: AXE is proposed as a flexible framework that infuses overflow avoidance guarantees into layerwise PTQ algorithms. It achieves this by controlling dot product ranges throughout the error correction process.

Theoretical Justification and Flexibility: The paper offers theoretical motivation for AXE and demonstrates its adaptability by implementing accumulator-aware variants of existing PTQ algorithms, GPFQ and OPTQ.

Support for Multi-Stage Accumulation: Unlike prior accumulator-aware QAT methods that assume a monolithic accumulator, AXE is designed to support multi-stage accumulation (e.g., tiled dot products). This opens the door to full datapath optimization for large language models (LLMs).

Improved Trade-off between Accumulator Bit Width and Model Quality: Experiments show that AXE significantly improves the trade-off between accumulator bit width and model quality compared to alternative methods like EP-init and naive bit width manipulation. This translates to lower power consumption with better model quality (Figure 1, Figure 4).

Scalability to Billion-Parameter LLMs: AXE demonstrates excellent scalability to billion-parameter LLMs. For instance, when quantizing Llama3 8B for a 16-bit multi-stage accumulation datapath, AXE maintains up to 98% of the FP16 perplexity, surpassing naive bit width manipulation by up to 15%. This is critical for wider transformer architectures.

Compatibility with Pre-processing Algorithms: The framework is shown to be compatible with established pre-processing methods like SmoothQuant and Hadamard-based incoherence processing (Table 2).

Analysis of Accumulator Bit-width Impact: The paper provides evidence that addition costs can dominate multiplication costs as operand precision is reduced, highlighting the growing importance of accumulator-aware quantization (Figure 2, Section 3).

**Audience:**

Yes

**Audience Explanation:**

Here's why, considering TMLR's scope:

Relevance to Core ML Research (Quantization): Quantization is a fundamental and active area of research in machine learning, particularly for deploying models efficiently on hardware. TMLR's audience, which includes researchers and practitioners working on model optimization, hardware-aware ML, and efficient inference, would naturally be interested in advancements in this field.

Addressing a Critical and Emerging Bottleneck (Accumulator-Awareness): The paper highlights that as operand bit widths shrink, the cost of additions in MAC units becomes a dominant bottleneck, potentially even more so than multiplications. This is a crucial insight that challenges traditional focus and points to a new frontier in quantization. Researchers striving for maximum efficiency would find this motivation highly relevant.

Focus on Large Language Models (LLMs): LLMs are currently at the forefront of AI research and application. Their massive size makes efficient deployment (especially on edge devices or with limited resources) a paramount concern. A framework that specifically addresses accumulator-aware quantization and scales to billion-parameter LLMs (like Llama3 8B) is directly relevant to a large segment of TMLR's audience working with these models.

Theoretical Guarantees (Overflow Avoidance): For many applications, particularly those requiring high reliability or safety, theoretical guarantees (like overflow avoidance) are extremely valuable. The paper's mathematical framework and explicit guarantees would appeal to researchers interested in robustness and correctness in ML systems.

**Broader Impact Concerns:**

While AXE prevents overflow-induced catastrophic failure, it doesn't guarantee full fidelity to the FP16 model's behavior in other aspects (e.g., subtle shifts in bias, reasoning quality degradation before catastrophic failure). The technical "guarantees" for overflow might lead deployers to over-trust the model's overall performance in these extreme low-precision settings, potentially leading to unfair or incorrect outcomes that are difficult to debug in an edge environment.

**Claims And Evidence:**

Yes

**Claims Explanation:**

The submission presents a strong case for AXE, providing a good balance of theoretical justification, empirical evidence, and clear explanations. The claims are generally well-supported.

Claim: AXE is the first accumulator-aware PTQ framework explicitly designed to endow overflow avoidance guarantees.
Evidence: The paper states "To the best of our knowledge, ours marks the first formal study of accumulator-aware PTQ, and the first solution to scale to modern LLMs" (Page 4). It also explicitly mentions that prior accumulator-aware research was limited to QAT (Page 1, Abstract; Page 2).

Claim: Prior accumulator-aware quantization was limited to QAT, and PTQ lacked a formal study/solution .
Evidence: The abstract and introduction explicitly state this limitation ("Accumulator-aware quantization research has so far only considered the quantization-aware training (QAT) paradigm...", "To the best of our knowledge, there has been no formal study that explores accumulator-aware quantization in the PTQ setting."). This is a claim about the state of prior research, and the paper implicitly supports it by not citing any PTQ accumulator-aware methods that achieve similar guarantees or scaling.

**Requested Changes:**

Discussion on Computational Overhead of AXE:
Issue: The paper focuses on the quality and hardware efficiency (power/area through bit operations and accumulator size). However, PTQ methods inherently have some computational cost during calibration. AXE introduces additional operations. Briefly discuss the computational overhead introduced by AXE's components during the PTQ calibration phase. Is it negligible compared to the base PTQ algorithms (GPFQ/OPTQ)? Are there any situations where it becomes significant? This would provide a more complete picture of the framework.

Briefly Discuss Other Metrics Beyond Perplexity for LLMs:
Issue: The evaluation primarily uses WikiText2 perplexity for model quality and zero-shot reasoning accuracy for Llama3. While perplexity is standard, LLM evaluation is increasingly complex. Acknowledge the broader landscape of LLM evaluation. The paper already mentions zero-shot reasoning tasks (Table 3 in Appendix B) which is good. A sentence stating that perplexity is a strong proxy for generative quality and a necessary first step, followed by the zero-shot task evaluation, would be sufficient. This shows awareness without adding extensive, potentially distracting, new results.

---

> ### Author Response · Authors · 2025-10-27
>
> Thank you for the thoughtful and constructive feedback. We are glad that you found the paper’s motivation, theoretical formulation, and empirical support for AXE convincing and relevant to TMLR’s audience.
>
> **On computational overhead:**
> We agree that clarifying calibration overhead is valuable. In practice, the additional cost introduced by AXE is negligible (within run-to-run variation of wall-clock time) and thus not easily visible in plots or tables. We now explicitly note this in Section 5 of the revised manuscript — thank you for highlighting this omission.
>
> **On evaluation metrics:**
> We appreciate the suggestion to acknowledge the broader landscape of LLM evaluation.  We now clarify that perplexity serves as a standard and sensitive proxy for generative quality, and acknowledge that evaluation of LLMs is increasingly complex beyond just zero-shot accuracy. We also move our zero-shot accuracy results to the main text and emphasize further evaluation in the impact statement.
>
> **On broader impact:**
> We recognize that AXE’s overflow guarantees do not imply complete fidelity to FP16 behavior. We clarify that this limitation is not unique to AXE — all compression methods (e.g., quantization, pruning, distillation) can shift model behavior in subtle ways. We have updated our impact statement accordingly.
>
> We thank you again for these constructive insights, which have helped strengthen the clarity and completeness of our work.

---

### Review · Reviewer_64Px · 2025-09-29

**Summary Of Contributions:**

This paper introduces AXE, the first accumulator-aware framework for post-training quantization (PTQ) of large language models (LLMs). Unlike prior work that focused on accumulator-aware quantization-aware training (QAT), AXE provides theoretical guarantees against overflow in PTQ, and supports multi-stage accumulation. The authors implement AXE on top of existing PTQ algorithms (GPFQ and OPTQ), showing consistent improvements in power–accuracy trade-offs across a range of models from GPT-2 to Llama3.

Strengths:
1. Addresses a real hardware bottleneck as the field moves toward 4-bit and sub-4-bit inference.
2. Strong theoretical grounding with clear hardware-aware constraints.
3. Scalability to billion-parameter LLMs, which is non-trivial.
4. Flexible design—AXE is algorithm-agnostic and demonstrated on two popular PTQ methods.

Weaknesses:
1. Heavy reliance on perplexity as the main evaluation metric; broader task-level evaluations could strengthen the empirical case.
2. Coverage of models is somewhat narrow and outdated (mostly GPT, OPT, Llama); evaluation on additional families (e.g., Qwen, Seed, etc.) would strengthen the generality claim.

**Audience:**

Yes

**Audience Explanation:**

This paper directly addresses a growing pain point: as quantization pushes into 4-bit and below, accumulation—not multiplication—becomes the performance and power bottleneck. Given the surge in LLM quantization research, this work bridges a critical gap between theory and practice.

**Broader Impact Concerns:**

No ethical risks are apparent.

**Claims And Evidence:**

Yes

**Claims Explanation:**

The paper’s central claim—that AXE enables safe, high-accuracy PTQ under low-precision accumulation—is well-supported by both theoretical analysis and comprehensive experiments.
1. The Pareto frontier analyses convincingly demonstrate AXE’s improvements over baselines such as naïve bit width manipulation and EP-init.
2. Results scale from small to large models and include both perplexity and zero-shot reasoning tasks, supporting the generality of the approach.

**Requested Changes:**

1. Add evaluation on downstream tasks beyond perplexity (i.e., more benchmarks).
2. Extend experiments to include additional model families (e.g., Qwen) to demonstrate broader applicability.

---

> ### Author Response · Authors · 2025-10-27
>
> Thank you for the positive and thoughtful feedback. We are encouraged that you found the theoretical grounding, scalability, and practical relevance of AXE compelling.
>
> Thank you for the point you raised regarding broader model coverage and evaluation. We primarily focused on perplexity because it is a standard and sensitive proxy for LLM evaluation; we now clarify this point in Section 5. We also move our zero-shot accuracy results to the main text and emphasize further evaluation in the impact statement. We note that our evaluation already spans diverse transformer families (GPT, OPT, SmolLM, Pythia, and Llama) and a wide parameter range (70M–12B) with each Pareto frontier requiring roughly 750 experiments per algorithm — over 2,000 runs per plot and more than 15,000 total experiments for the LLM analyses alone. That said, we appreciate your request, so we have expanded the study. We now include image classification models (ResNet18, MobileNetV2, ViT), adding roughly another 5,000 experiments to demonstrate AXE’s generality beyond language models and beyond perplexity and zero-shot accuracy as an evaluation metric.
>
> We thank you again for recognizing the rigor and relevance of AXE and for feedback that helped clarify its scope and applicability.

---

### Review · Reviewer_cpM1 · 2025-10-04

**Summary Of Contributions:**

The paper introduces **AXE**, the first framework for accumulator-aware post-training quantization (PTQ) of large language models (LLMs). Prior work in this area was limited to quantization-aware training (QAT), which requires costly retraining or fine-tuning. AXE provides a theoretically justified method that guarantees overflow avoidance during low-precision accumulation, making it feasible to optimize not just multipliers but also adders in MAC units.

Key contributions include:

* **Formalization of accumulator-aware PTQ** and derivation of constraints ensuring overflow avoidance.
* **General framework (AXE)** applied to existing PTQ algorithms (GPFQ, OPTQ), shown to work with multi-stage accumulation.
* **Experimental validation** on models from GPT-2 to LLaMA-3 (up to 8B parameters), showing AXE preserves up to **98% FP16 perplexity** with significant power/bit-operation savings.
* **Scalability analysis**, demonstrating improved preservation of model quality as model size increases.

**Strengths**:

* Novelty: First systematic treatment of accumulator-aware PTQ.
* Theoretical rigor: Overflow-avoidance constraints derived and integrated into greedy PTQ algorithms.
* Strong empirical validation across multiple model families, scaling to billions of parameters.
* Practical impact: Multi-stage accumulation enables real hardware benefits (power, throughput).

**Weaknesses**:

* Limited evaluation of real-world latency/power on actual hardware platforms (results rely mainly on proxy cost models such as bit-flip counts).
* Focuses mostly on perplexity and reasoning benchmarks; does not deeply investigate downstream fine-tuned tasks.
* Comparisons are mainly against EP-init and naïve bit-width manipulation; stronger baselines (e.g., very recent PTQ methods) could strengthen claims.

**Audience:**

Yes

**Audience Explanation:**

TMLR’s audience includes researchers in machine learning efficiency, model compression, and LLM deployment. This work directly addresses a **pressing bottleneck in LLM quantization**, the under-explored role of low-precision accumulation. By bridging QAT and PTQ in this domain, the findings are highly relevant for both theory-focused researchers and practitioners seeking scalable deployment of LLMs.

**Broader Impact Concerns:**

The broader impacts are mostly positive: more efficient quantization methods can reduce energy consumption and enable deployment of LLMs on resource-constrained devices, improving accessibility.

Potential concerns include:

* **Environmental impact framing**: While AXE reduces energy at inference, the work does not address training costs or lifecycle impacts.
* **Misuse potential**: As with any efficiency method, lowering barriers to deploying LLMs may accelerate the proliferation of models in harmful applications.

The paper could benefit from a more explicit **Broader Impact Statement** addressing these points.

**Claims And Evidence:**

Yes

**Claims Explanation:**

The paper supports its claims with both **theoretical analysis** (derivation of constraints and guarantees) and **extensive experiments** across a wide range of models and settings. The claims about Pareto dominance (better perplexity vs. accumulator width) are well-supported by multiple figures and tables. The extension to multi-stage accumulation is motivated theoretically and confirmed empirically. The main caveat is that the evaluation of power/latency benefits is indirect (via cost models), but the evidence is still clear and convincing for the quantization/ML research community.

**Requested Changes:**

* **Critical**: Provide more discussion of **real-world hardware implications**. The paper currently relies on cost models and simulation (bit-flip counts) but does not benchmark actual inference throughput/latency/power. Even small-scale hardware experiments would significantly strengthen claims.
* **Critical**: Expand comparison baselines. Including recent PTQ methods beyond EP-init and naïve bit-width manipulation would provide a fairer picture of where AXE stands in the current literature.
* **Optional**: Explore additional downstream tasks (beyond perplexity and zero-shot reasoning) to evaluate robustness of AXE-quantized models.
* **Optional**: Improve clarity by summarizing design trade-offs (weights/activations vs. accumulator precision) in a concise table for practitioners.

---

> ### Author Response · Authors · 2025-10-27
>
> Thank you for the detailed and constructive feedback. We are encouraged that AXE’s novelty, theoretical rigor, and scalability were well-received, and we address the specific comments below.
>
> **On real-hardware implications:**
> Thank you for bringing this up and giving us the opportunity to clarify. As we note in the paper, modern hardware supporting low-precision accumulation is not yet widely available, making direct latency or power measurements infeasible. We agree that connecting to hardware measurements is important. In fact, our cost analysis is derived from gate-level RTL simulations of realistic MAC datapaths, which faithfully capture switching activity and accumulator bit width effects. We are sorry if this was not clear from our discussion. We now clarify this limitation in Section 3 and explicitly explain that our methodology reflects the most accurate available proxy.
>
> **On comparison baselines:**
> We appreciate the request for broader comparisons. However, to our knowledge, AXE is the first accumulator-aware PTQ framework; prior work addressed this problem only in the quantization-aware training (QAT) setting.
>
> **On evaluation scope and clarity:**
> We now expand our discussion in Section 5 acknowledging the broader evaluation landscape (e.g., beyond perplexity), we move our zero-shot accuracy results to the main text to emphasize its importance, and we now include an additional 5,000 experiments with image classification models to demonstrate AXE's generality beyond language models. Additionally, please note that the design trade-offs between weight, activation, and accumulator precision are  summarized in the final appendix tables.
>
> We thank you again for your thoughtful and helpful feedback.

---

### Review · Reviewer_i5PD · 2025-10-07

**Summary Of Contributions:**

The authors propose a post-training quantization (PTQ) framework that explicitly incorporates accumulator-aware constraints to prevent overflow, providing theoretical guarantees absent from prior work.


### Strengths

1. Solid theoretical foundation with explicit hardware constraints
1. Scalability evaluation shows quality maintained for larger models.


### Weaknesses

1. Evaluation relies heavily on perplexity and limited tasks.
1. Experiments cover a narrow set of models.

**Audience:**

Yes

**Audience Explanation:**

By introducing accumulator-aware PTQ with theoretical guarantees, the paper addresses a foundational problem that connects algorithmic quantization design with hardware-level implications.

**Claims And Evidence:**

Yes

**Claims Explanation:**

The computational efficiency gains are theoretically proven and are numerically demonstrated for different architectures: Llama3 and SmolLM2.

The modularity of the method is demonstrated through implementation on top of two algorithms: GPFQ and OPTQ.

**Requested Changes:**

1. Explicitly state which aspects of AXE are original contributions versus adaptations of existing accumulator-aware or bounded-range quantization ideas.

2. The paper should discuss the computational overhead of AXE during PTQ calibration.

3. Add more benchmarks.

---

> ### Author Response · Authors · 2025-10-27
>
> Thank you for the constructive feedback and for recognizing the theoretical and practical contributions of AXE.
>
> **On explicitly summarizing contributions:**
> Thank you for this suggestion. We now explicitly delineate our contributions as a subsection in the introduction.
>
> **On computational overhead:**
> We agree that clarifying calibration overhead is valuable. In practice, the additional cost introduced by AXE is negligible (within run-to-run variation of wall-clock time) and thus not easily visible in plots or tables. We now explicitly note this in Section 5 of the revised manuscript — thank you for highlighting this omission.
>
> **On benchmarks:**
> We now expand our discussion in Section 5 acknowledging the broader evaluation landscape (e.g., beyond perplexity), we move our zero-shot accuracy results to the main text to emphasize its importance, and we now include roughly 5,000 additional experiments with image classification models to demonstrate AXE’s generality beyond language models.
>
> We thank you again for the thoughtful comments and constructive suggestions.

---

### Decision · Action_Editor_tKAX · 2025-11-24

**Recommendation:** Accept with minor revision

**Additional Comments:**

Overall, this is a solid and well-written paper. The main ideas are clear, the theory is clean, and the experiments are thorough enough to support the conclusions. For the minor revision, I encourage the authors to address the points raised repeatedly by reviewers: briefly clarifying calibration overhead, making the contribution boundaries a bit more explicit, and acknowledging the broader evaluation landscape for LLMs. These should be straightforward to incorporate. I do not recommend a certification.

**Audience:**

Yes

**Audience Explanation:**

The topic sits squarely within the growing interest in efficient LLM deployment and hardware-aware quantization. As models move further into low-precision regimes, accumulator behavior has become increasingly important, and this paper provides a timely and well-motivated treatment of that issue. I expect that researchers working on compression, systems-ML, and practical LLM inference will find the work relevant.

**Claims And Evidence:**

Yes

**Claims Explanation:**

The paper presents a clear formulation of accumulator-aware PTQ and supports it with both theory and a fairly extensive set of experiments. The overflow-avoidance constraints are well explained, and the empirical results across several model families are consistent with the claims. The two main gaps noted by reviewers—no real hardware measurements and a somewhat narrow set of baselines—are valid, but they do not undermine the core technical contribution.

---

> ### Author Response · Authors · 2025-12-05
>
> We sincerely thank the Action Editor for their guidance throughout the review process. We incorporated all the changes requested from the reviewers in the previous revision and have updated the final camera-ready version to include references to our open-source implementations. We believe this final version reflects all the improvements suggested by the Action Editor and the reviewers.